# Electrifying passenger road transport in India requires near-term electricity grid decarbonisation

Amir F. N. Abdul-Manan [1,2✉], Victor Gordillo Zavaleta [3], Avinash Kumar Agarwal[4], Gautam Kalghatgi [5] & Amer A. Amer[2]

Battery-electric vehicles (BEV) have emerged as a favoured technology solution to mitigate transport greenhouse gas (GHG) emissions in many non-Annex 1 countries, including India. GHG mitigation potentials of electric 4-wheelers in India depend critically on when and where they are charged: 40% reduction in the north-eastern states and more than 15% increase in the eastern/western regions today, with higher overall GHGs emitted when charged overnight and in the summer. Self-charging gasoline-electric hybrids can lead to 33% GHG reductions, though they haven't been fully considered a mitigation option in India. Electric 2-wheelers can already enable a 20% reduction in GHG emissions given their small battery size and superior efficiency. India's electrification plan demands up to 125GWh of annual battery capacities by 2030, nearly 10% of projected worldwide productions. India requires a phased electrification with a near-term focus on 2-wheelers and a clear trajectory to phase-out coal-power for an organised mobility transition.

[1] Strategic Transport Analysis Team, Beijing Research Center, Aramco Asia, Beijing, China. [2] Transport Technologies R&D Division, Saudi Aramco Research & Development Center (R&DC), Dhahran, Saudi Arabia. [3] Aramco Fuel Research Center, Aramco Overseas Company B.V, Paris, France. [4] Engine Research Laboratory, Department of Mechanical Engineering, Indian Institute of Technology Kanpur, Kanpur, India. [5] Consultant Professor, Shanghai Jiao Tong University, Shanghai, China. ✉email: amir.abdulmanan@aramco.com

Transport was responsible for a quarter of global $CO_2$ emissions from fuel combustion[1], and light-duty passenger road vehicles (LDV) accounted for around 40% of transport emissions in 2018[2]. This sector's key mitigation responses can be structured using the 'avoid-shift-improve' framework[3,4]: avoiding unnecessary motorised journeys and shifting to lower carbon transport modality while improving fuel and powertrain technologies. Despite the need for a wide range of policies[5], climate change mitigation strategies are often technology-centric. In the Paris Agreement, more than 65% of the transport mitigation measures proposed in the Nationally Determined Contributions (NDC) have a strong technology focus[3]. Electrification of transport is a good example of a technological fix that has been the narrow focus of many government policies and mandates at the expense of a more comprehensive approach that is much needed[4,5].

Electrification of LDVs shifts the use phase greenhouse gas (GHG) emissions from vehicle tailpipe to electricity generation. Therefore, the power generation mix must be adequately decarbonised to enable deep reductions in GHG emissions from the passenger road LDV sector. In some geographical regions with high clean and renewable power penetration, battery-electric vehicles (BEVs) offer a significant reduction in transport GHGs[6–10]. Diversity in power generation profiles and differences in fleet characteristics[5] implies that the BEVs' GHG reduction potentials cannot be generalised worldwide[6,11,12]. Still, many non-Annex 1 countries intend to promote the uptake of BEVs, in which the benefits are yet to be adequately quantified within the specific national context. The EV30@30 campaign, under the Electric Vehicle Initiatives (EVI) of the Clean Energy Ministerial, aims for BEVs to account for 30% of new vehicle sales by 2030[13]. As one of the participating countries, India recently launched phase 2 of the Faster Adoption and Manufacturing of Electric Vehicles (FAME II) scheme, which provides 1.3 billion USD fiscal incentives over the next three years to support the rapid electrification of the transport sector[14].

After the power generation and industrial sectors, India's transport sector is the third-largest energy-related GHG emission source accounting for about 13% in 2019[15]. Between 2000 and 2019, transport energy demand in India grew 3.5 times, and GHG emissions from passenger road transport alone quadrupled[15]. With rising disposable income, India has experienced a five-fold growth in per capita vehicle ownership, registering a total road vehicle stock of over 250 million in 2019 comprising of two-wheelers (2 W), three-wheelers (3 W), and four-wheelers (4 W)[15]. Vehicle ownership in India is geographically diverse. It can be more than 100 cars per 1000 population in large cities like Delhi, Chennai, and Coimbatore[16]; however, it has a more modest national average of only 37 cars per 1000 population in 2019[15].

In contrast, other G20 countries like the USA, Japan, and Europe have car ownership rates substantially higher than 500 cars per 1000 population[17]. With the expected growth in per capita GDP, India will likely go down the same path unless effective mitigation responses are deployed early. With a population of more than 1.3 billion, accounting for 17.8% of the global population in 2019, a lack of broad policy interventions encompassing the 'avoid-shift-improve' framework can significantly increase GHG emissions from the Indian transport sector. Thus, any successful effort to limit the global temperature increase to below 1.5 °C will have to consider India's growing appetite for mobility and private vehicle ownership.

Here, we model the life cycle GHG emissions of over 600 passenger road vehicles comprising two-wheelers (2 W) and four-wheelers (4 W) available in India in the financial year 2018/19. This exercise incorporated emissions from the energy life cycle (i.e., fuels and electricity production), vehicle manufacturing (including battery production and end-of-life recycling), and vehicle use. The assumptions made in this study are in the Methods section and are supplemented by supporting documents. We examine the implications of regional heterogeneity within the vast and diverse Indian subcontinent, including ambient temperatures, seasonal variations, and daily fluctuations in power generation profiles on the life cycle GHG emissions of commercially available electric 2-wheelers and 4-wheelers. Next, we assess the life cycle GHG emission ranges of passenger road vehicles in India by powertrain types. The approach aims to identify technologies enabling large GHG reductions within the road transport sector. The electrification of 2-wheelers presents a big opportunity to reduce GHG emissions. An electric 2 W is substantially more energy-efficient than a traditional gasoline 2 W, and it is equipped with a small battery pack; therefore, it carries only a minor manufacturing GHG penalty. However, the climate change mitigation potential of 4 W BEVs is critically dependent on how they are charged[6–8,11]. The time of the day and the year when the BEV is charged and the region where the charging occurs considerably affect its overall life cycle GHG emissions. We conclude that electrification can offer climate change mitigation potential in some sectors and regions, but it is not the panacea for India's passenger road transport. Climate change responses for the transport sector necessitate a balanced mix of behavioural modifications and technology-oriented policies with diversity across the three dimensions - time, region, and sectors. Although climate change is a global challenge, there is no universal, one-size-fits-all technology solution and policy solution. Ultimately, India's EV30@30 ambition must be complemented by a near-term commitment to phase-out coal from the power sector to improve the GHG reduction prospect of a 4 W BEV. In the absence of a clear trajectory to phase down coal, the electrification of transport should prioritise India's growing 2-wheel segment that could still offer about a 20% GHG benefit. This has to be complemented by a more rigorous fuel efficiency standard and low-carbon fuels standard to drive the adoption of highly-efficient 4 W engines (e.g., hybridised vehicles) and fuels with a lower climate impact (e.g., sustainable low-carbon fuels) as a transitional solution.

## Results

**When and where BEVs are charged.** There are a limited number of commercially available 4 W BEVs in India. In 2019, BEVs represented only 0.1% of total 4 W passenger LDV sales. We were able to identify three vehicle models (i.e., Tata Nexon, Mahindra Verito, and Tata Tigor) with three different powertrain variants but comparable performance, size, and weight: gasoline spark-ignition, diesel compression-ignition, and battery-electric (Table 1). These three models are compared for their life cycle GHG emissions.

In certain but not all geographical regions (Fig. 1), some BEVs can offer lower GHG emissions on a life cycle basis, though it also depends on the time of day when the charging takes place (Fig. 2). The climate change mitigation potential of a BEV is particularly diminished compared to a diesel vehicle, given that the latter has ~25% lower fuel consumption than an equivalent gasoline vehicle (Table 1). There is a marked benefit for BEVs in the north-eastern region because the power sector is mainly based on natural gas and hydroelectric. However, the north-eastern region accounts for only 3.7% of India's population in 2011[18], contributed less than 5% to the national GDP in 2018/2019[19], and accounts for about 2% of registered vehicles in 2015[20].

Overall, the Indian subcontinent is still heavily dependent on coal, which accounted for more than 70% of the total electricity generated during 2018/2019[21]; in the eastern and western provinces, however, the shares of coal-generated electricity were much higher at 89% and 84%, respectively (Supplementary Fig. 3). Renewable electricity generation has more than doubled

**Table 1 Summary of key vehicle characteristics for the 2-wheelers and 4-wheelers.**

| 4-wheelers | curb weight (kg) | wheelbase (mm) | Battery size (kWh) | ARAI-certified based on MIDC* | | |
| --- | --- | --- | --- | --- | --- | --- |
| | | | | Electric range (km) | Energy consumption (Wh/km) | Fuel consumption L/100 km |
| Tata Nexon | | | | | | |
| *Electric* | 1400 | 2498 | 30.2 | 312 | 100 | – |
| *Gasoline 1.2 L* | 1237 | 2498 | – | | – | 5.6 |
| *Diesel 1.5 L* | 1305 | 2498 | – | – | – | 4.2 |
| Mahindra Verito | | | | | | |
| *Electric* | 1225 | 2630 | 21.2 | 110 | 164 | – |
| *Gasoline 1.4 L* | 1080 | 2630 | – | – | – | 7.4 |
| *Diesel 1.5 L* | 1140 | 2630 | – | – | – | 4.8 |
| Tata Tigor | | | | | | |
| *Electric* | 1590 | 2450 | 21.5 | 213 | 101 | – |
| *Gasoline 1.2 L* | 1100 | 2450 | – | – | – | 4.9 |
| *Diesel 1.05 L* | 1130 | 2450 | – | – | – | 4.1 |
| **2-wheelers** | **curb weight (kg)** | **Fuel Consumption (L/100 km)** | **Battery size (kWh)** | **Electric range (km)** | **Energy consumption (Wh/km)** | **Maximum speed (km/h)** |
| Gasoline 2 W (Class 1) | | | | | | |
| India Average | 109 | 1.7 | – | – | – | NA |
| Electric 2 W (Class 1) | | | | | | |
| Revolt RV400 | 108 | – | 3.2 | 156 | 21 | 80 |
| TVS iQube | 118 | – | 2.3 | 75 | 30 | 78 |
| Ather 450 Plus | 111 | – | 2.7 | 107 | 25 | 80 |
| Hero Optima e5 | 73 | – | 1.5 | 82 | 19 | 42 |
| Hero Nyx e5 | 77 | – | 1.3 | 50 | 27 | 40 |
| Hero Photon LP | 87 | – | 1.9 | 80 | 23 | 45 |
| Hero Flash | 87 | – | 1.0 | 50 | 19 | 25 |
| Hero Flash e2 | 69 | – | 1.5 | 85 | 18 | 25 |
| Hero Optima | 86 | – | 1.0 | 50 | 19 | 25 |
| Hero Optima e2 | 68 | – | 1.5 | 85 | 18 | 25 |
| Hero Optima ER | 83 | – | 1.5 | 122 | 13 | 42 |
| Hero Nyx ER | 87 | – | 2.7 | 100 | 27 | 42 |
| Okinawa Lite | 96 | – | 1.3 | 60 | 21 | 25 |
| Okinawa Ridge+ | 96 | – | 1.7 | 84 | 21 | 45 |
| Okinawa iPraise+ | 96 | – | 3.3 | 139 | 24 | 58 |
| Okinawa PraisePro | 96 | – | 2.0 | 88 | 23 | 58 |

*ARAI Automotive Research Association of India, MIDC Modified Indian Drive Cycle
Data collected from the manufacturers' websites in the year 2020.

between 2014 and 2019[22], resulting in combined wind and solar power generation of 101 TWh, which provided only 7% of India's total 1376 TWh of electricity demand during the same period.

Moreover, solar and wind power availability is not consistent throughout the day. Solar irradiation tends to peak mid-day, causing the hourly power-generation emission intensity to follow a duck curve, with its highest intensities achieved overnight (Supplementary Fig. 4(c)). The intermittencies of renewables mean that the GHG emission profiles of BEVs also vary with time (Fig. 2), reflecting the hourly and seasonal fluctuations in power generation mixes.

The life cycle emissions of BEVs oscillate over a wide range depending on when the charging takes place (Fig. 2 and Supplementary Fig. 4). Off-peak, overnight charging in India results in larger GHG emissions than daytime charging—a difference of 3-9%. Surveys in other countries indicate that BEV users have a strong preference to charge at the convenience of their homes and during the night[23,24]—as reported by respondents in Korea[25], Netherlands[26], USA[27], UK[28,29] and Norway[30]. It is unclear how many private motorists in India would have access to home charging, but a time-differentiated electricity tariff[31] can encourage lower emissions charging behaviour.

**Cleaner power outlook.** By 2030, the shift towards renewables and lower reliance on coal power plants would mean that every unit of electricity produced has, on average, 27% lower GHG

emissions compared to the year 2018/19 (Supplementary Fig. 3). Correspondingly, it allows BEVs in 2030 to reduce their life cycle emissions by 23–25% (Supplementary Fig. 15) relative to a BEV in 2018/2019. A more rapid reduction in the electric grid carbon intensity could enable a larger reduction in the life cycle GHG emissions of the BEV. Additionally, the climate change mitigation potential for a BEV in 2030 also depends on its electricity consumption for every kilometer driven (i.e., Wh/km) and the fuel efficiency of the combustion engine vehicle baseline (i.e., L/100 km) for that year (Fig. 3). Phase 2 of the fuel consumption standard in India, which comes into effect from the year 2022/2023 onwards[32], stipulates that 4 W passenger vehicles with an average curb weight of 1237 kg should not exceed 4.95 L/100 km. Its overall GHG emissions would be comparable to a BEV with a 142 Wh/km (Fig. 3) for a gasoline vehicle. To put this into perspective, in 2018, the industry average energy consumption of BEVs in China was 143 Wh/km[33]. A BEV less efficient would have higher overall GHG emissions than a comparable gasoline 4 W vehicle. On the other hand, a more efficient BEV would produce lower overall GHG emissions. Hence, strategies to electrify future passenger transport in India must involve developing energy-efficient BEVs and deep decarbonisation of the power sector to ensure an overall reduction in GHG emissions.

Limiting the global temperature increase to below 1.5 °C will require almost complete decarbonisation of the power sector by

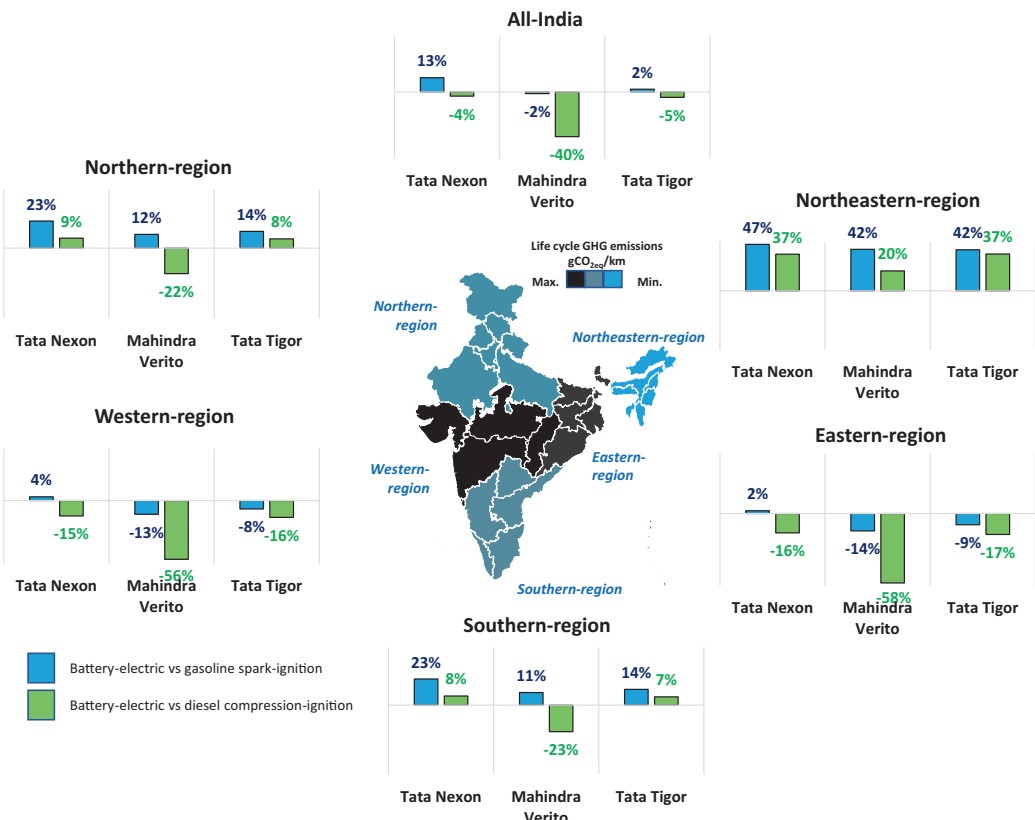

**Fig. 1 GHG emission reduction potential of electric 4-wheelers in India.** A positive value denotes the percentage emission reduction, while a negative value means that the BEV emits higher GHG than its gasoline/diesel counterparts. The life cycle GHG emissions of BEVs in India vary significantly based on individual regions' power generation profiles, with the generation mix assumed to be static throughout the vehicle lifetime. There are larger overall GHG emission benefits for BEVs in the north-eastern region and to a lesser extent in the northern and southern regions. The climate change mitigation potential of a BEV is diminished when contrasted against diesel engines. Supplementary Fig. 8 provides additional details.

the mid-century. Progress towards this target will lower emissions of BEV, resulting in substantial GHG benefits for the passenger road transport sector (Fig. 4). However, India has the dual challenge of decarbonising the grid while ensuring reliable and affordable electricity nationwide. Electricity consumption in India varies from a low of 311 kWh per capita in Bihar to a high of 2378 kWh per capita in Gujarat[15]. The per capita electricity consumption in high-income cities in India (e.g., Delhi) is still only about half of the global average[15]. Therefore, there is a significant electricity demand growth prospect. Despite achieving near-universal household grid connectivity in 2019, more than 50% of Indian households still report multiple power cuts daily[15]. Therefore, India's rate of transport electrification should also consider its progress towards nationwide access to a reliable, affordable, and sustainable electricity supply.

**Electrified solution**. More efficient LDVs, such as the gasoline-electric and diesel-electric hybrids, have entered the Indian market. Compared to traditional gasoline 4-wheelers, these vehicles reduce emissions by 21%, 25%, and 33% for diesel, hybrid diesel, and hybrid gasoline, respectively (Fig. 5). Moreover, no additional infrastructure investment is required for the benefits to materialise. Thus, it presents an opportunity to accelerate the mitigation of GHGs in India's 4 W LDV sector. In contrast, BEVs' real GHG mitigation potential will critically rely on the speed of grid decarbonisation through investments in cleaner power and the widespread deployment of charging networks. Given the power mix in India in 2019, BEVs offer smaller GHG reduction potential and higher uncertainty due to geographical

and temporal variabilities (Supplementary Fig. 9). It is, therefore, prudent for policies to also encourage rapid development and uptake of advanced hybrid solutions with varying electrification levels.

Presently, hybrid vehicles in India are either micro or mild hybrids, with much smaller battery sizes, typically below 2kWh. The other end of the electrification spectrum is pure battery-electric vehicles equipped with much larger batteries: in the Indian market, the typical range is between 20 and 30 kWh. However, it is important to note that the global trend is towards larger battery sizes, given that the automakers are racing for a longer all-electric range. China, which accounts for 55% of global BEV sales[34], has an average battery size of 37kWh in 2018, reflecting an increase of 32% and 60% compared to 2017 and 2014, respectively[33]. The installed battery capacity is substantially bigger for larger passenger car segments. The average battery capacity for the B-segment car in China is 60kWh, 62% above the industry average[33]. This is particularly relevant given a structural shift towards larger passenger cars. Many countries are experiencing sharply rising Sport Utility Vehicles (SUV) sales, reaching 50%, 44%, 36%, and 34% of total car sales in 2019 in the USA, China, Europe, and India, respectively[35].

**Electric future rides on 2-wheels**. 26.2 million vehicles were sold in India in 2018/19, and the vast majority were 2-wheelers like scooters, motorcycles, and mopeds with a combined market share of over 80%[36]. Generally, a 2 W vehicle is about 70% less GHG intensive than a 4 W vehicle on a gCO$_{2eq}$/km basis (Supplementary Figure 21). Still, cumulatively, Indian 2-wheelers account

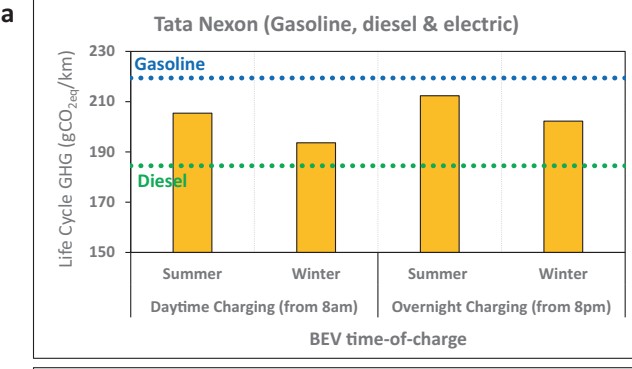

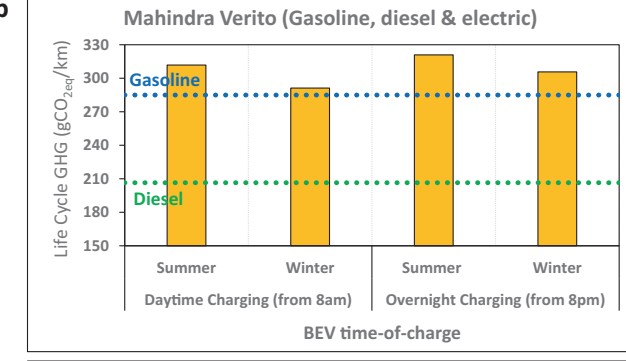

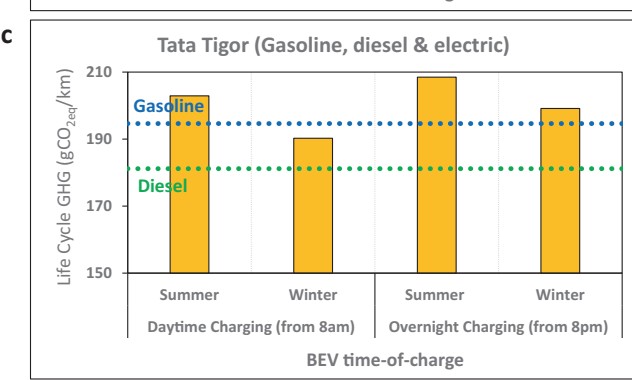

**Fig. 2 Seasonal and diurnal variations.** Effects of battery recharging time on the life cycle GHG emissions of battery-electric vehicles (BEV) based on the 24 h grid fluctuations of a typical day in winter and summer in India in 2018/2019 (refer to Supplementary Fig. 4 for the hourly grid fluctuations in India and the impacts on the carbon intensity of electricity). Daytime charging starts at 8 am, while overnight charging starts at 8 pm. Charging duration is estimated based on the BEV's battery size and the use of a typical 2.5 kW charger [102], with a total vehicle lifetime distance of 200,000 km for (**a**) Tata Nexon, (**b**) Mahindra Verito, and (**c**) Tata Tigor.

for as much emissions as 4-wheel passenger cars (i.e., 18% of total transport GHGs) because of their substantial fleet size[15]. In 2019, out of 255 million passenger vehicles on the road in India, only 51 million were 4 W vehicles. Therefore, unlike other countries, mitigating GHGs emitted by 2-wheelers would be key to addressing India's growing transport emissions.

The electrification of 2-wheelers presents an enormous potential for reducing GHG emissions (Fig. 6b) and improving air quality, particularly in densely populated cities[37]. An electric 2-wheeler is equipped with a smaller battery size, which means that its total vehicle manufacturing emissions is not significantly higher than a conventional gasoline 2-wheeler (Supplementary Figure 21). Combined with its much higher powertrain efficiency, an electric 2-wheeler has an overall GHG advantage over the traditional 2-wheelers.

The median gasoline consumption of traditional 2 W in India is 2 L/100 km (Fig. 6a)[38], which is within the average range reported by Anup & Yang[39]. Electric 2 W are primarily in Class-1 with a maximum of 150 km/h. Class-1 gasoline 2 W has a median energy consumption of 15kWh/100 km, while an electric 2 W in India typically consumes about 2kWh/100 km (Table 1). The life cycle GHG emissions of electric 2 W vary across the Indian subcontinent due to the diverse power generation mixes. However, on average, it is still 20% lower than the median gasoline 2-wheeler (Fig. 6b).

Compared to 4-wheelers, 2-wheelers are driven at lower speeds (averaging at 25 km/h[40]) and shorter distances (averaging about 7 km in the urban areas[15]). This makes e-scooters more energy-efficient for city commute and suitably equipped with a much smaller battery capacity. The battery can be easily removed and recharged indoors or swapped at a battery interchange station[41,42]. This eliminates the need for expensive charging infrastructure creation and long charging times, common drawbacks associated with 4 W BEVs.

**Criticality of sustainable battery materials.** India's annual new vehicle sales are projected to exceed 51.2 million vehicles by 2030[43]. India has pledged to achieve 30% EV sales under the EV30@30 campaign, which translates to annual sales of about 2 million new battery-electric passenger cars and over 12.7 million electric 2-wheelers in 2030[43]. The electric fleet in India could need up to 125 GWh of annual battery manufacturing capacity by 2030 (Fig. 7), with over 65% of the demand due to electric 4 W. Assuming a conservative battery size of 30–40kWh, the passenger car sector in India alone would require 63–83 GWh of batteries annually, which is 3–4 times the size of current production at Tesla's Gigafactory-1 in Nevada[44], or about 6% of the total worldwide lithium-battery production capacity projected for 2030[45,46]. This has important implications for future supplies of sustainably produced, critical minerals used in lithium-ion battery manufacturing, and India's ability to meet domestic growth in demand. India and other countries will have to compete against China's large domestic battery demand and dominant control over 80% of the global raw material refining, 77% of the global cell manufacturing capacity, and 60% of the global component manufacturing capacity[47].

Lithium-ion batteries heavily depend on several critical metals. Lithium, nickel, and cobalt are typical cathode materials, while graphite is used as the anode material. The geographical distributions of these metal reserves are uneven. The US Geological Survey (USGS) estimated that two mining countries accounted for 70% of each, lithium (Chile and Australia), cobalt (DR Congo and Australia), and graphite (China and Brazil) supply worldwide[48]. Concerns have been raised that the absence of international governance on critical resources could impede global progress towards the wider United Nations (UN) Sustainable Development Goals (SDG)[49], and it is particularly detrimental to vulnerable mining countries[50]. Although terrestrial mining will likely dominate supplies of critical metals in the medium term, the International Seabed Authority, an organisation under the UN Convention on the Law of the Sea, is currently drafting regulations relating to seabed mining[51]. While the prospects of oceanic mining of minerals could alleviate the supply constraints worldwide[51], the process needs to be science-based to ensure environmental safeguards, which should be put in place to mitigate the risk of accidental damage to the fragile ecosystem in the deep-seas[52,53].

Given the centrality of metal resources in the low-carbon transition, India must incorporate critical metals into its energy and climate plans, emphasising resource-use efficiency, recycling,

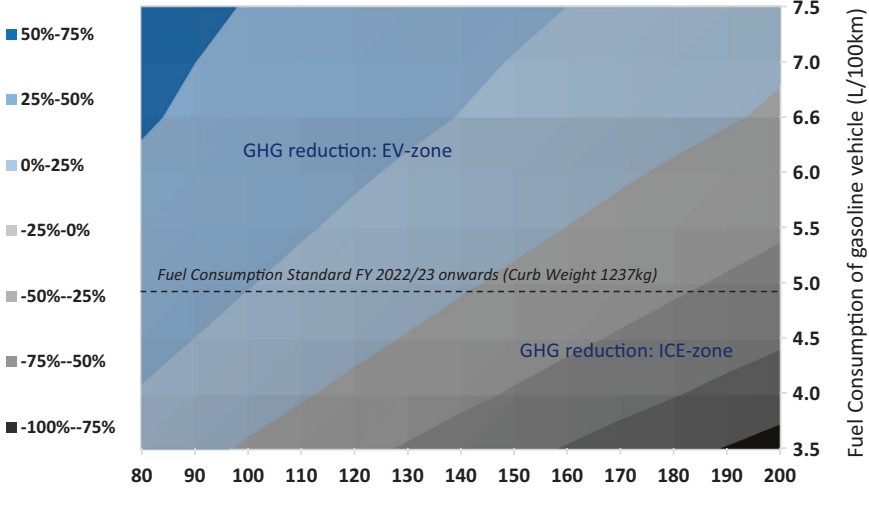

**Fig. 3 Contour plot mapping the life cycle GHG emission reduction (in percentage) of battery-electric vehicles (BEVs) with a cleaner power future.** The availability of cleaner electricity in 2030 can enable larger GHG reduction for BEVs. The percent GHG reduction depends on the efficiency of future BEVs and the baseline gasoline vehicle in 2030. The energy generation mix in 2030 is based on the national annual average. It is assumed to be static, with additional sensitivity plots in Supplementary Fig. 15 (**b**) of the supporting document and Fig. 4 below. [EV-zone = electric vehicle (EV) results in a lower overall GHG emission; ICE-zone = internal combustion engine (ICE) results in a lower overall GHG emission].

and repurposing battery materials[51,54,55]. Self-charging hybrid vehicles with varying electrification levels demand less critical raw materials because of their smaller battery capacity[9,56]. Thus, it can be an important bridge for effective GHG mitigation until the grid is sufficiently decarbonised. Battery-supply constraints would gradually ease – either by breakthroughs in new battery chemistries[57,58] or by supply-side investments having caught up with the demand. Sustainable mining and production of the battery have to become the industry norm.

**Miscounting carbon.** The Indian fuel consumption standard is based on the corporate-average method. Automakers may sell individual models that are better or worse than the specific fuel consumption targets determined by the vehicle weight, as long as the automaker's overall sales-weighted average fuel consumption meets the regulatory standard. The fuel consumption is derived from traditional combustion vehicles' tailpipe emission measured in $gCO_2$/km over the New European Driving Cycle (NEDC). Meanwhile, for a BEV, the gasoline-equivalent fuel consumption value is derived from its electricity consumption (Wh/km) based on gasoline's equivalent volumetric energy content.

Fuel economy standard is an important regulatory component driving efficiency improvements; however, for it to double as a climate change mitigation tool, existing standards must be strengthened to plug emissions loopholes that can weaken overall policy effectiveness[4,59]. The fuel economy standard in India, like most other countries, is based on tailpipe emissions. Therefore it provides preferential treatment for vehicles with zero exhaust emissions beyond their overall GHG benefit[59,60].

For example, under India's fuel economy standard, a BEV with a curb weight of 1237 kg that consumes 100 Wh/km would have a gasoline-equivalent fuel consumption of only 1.0 L/100 km. Crucially, this is 3.5 and 4.7 times lower than the actual gasoline-equivalent fuel consumption when calculated based on its overall life cycle GHG emissions in 2030 and 2019, respectively. For the BEV to have a low gasoline-equivalent fuel consumption value of 1.0 L/100 km, as prescribed within the regulation, the power sector will have to decarbonise substantially, achieving a net carbon intensity of no more than 150

$gCO_{2eq}$/kWh (Fig. 8), which is 78% below the 2030 forecast for electricity in India.

Most fuel economy and vehicle emission standards worldwide exempt the GHG consequences of upstream emissions, including power generation. While several preferential treatments were originally designed into regulations to jump-start the BEV industry[60–62], its continued and widespread use risks undermining efforts to mitigate GHG emissions from road transport globally[59]. A recent study conducted by the US Argonne National Laboratory estimated that existing loopholes in the transport regulations in the US, EU, and China lead to more than 1 billion tonnes of unaccounted $CO_{2eq}$ within the 2012-2025 timeframe, with the omitted upstream electricity emissions alone responsible for over 400 million tonnes of $CO_{2eq}$[59]. Miscounting carbon in a regulation designed to combat climate change is a serious error. Policymakers should consider a more comprehensive emissions framework, such as the life cycle assessment, to minimise leakages outside the regulated sectors. In this regard, automotive regulations are lagging behind fuel regulations[63]. The latter has evolved to incorporate life cycle assessment approaches within several regulatory instruments such as the low carbon fuels standards and the renewable fuels directives in some jurisdictions. It is imperative to mitigate these gaps by carefully strengthening the regulations and integrating with other policies in the mix to achieve a deeper and steeper emission reduction.

**Reflecting on the limits of generalisations.** Like any other modelling exercise, the strength of our analysis depends on the methodological approach and assumptions made. Simplifying conclusions, either for or against electrifying India's passenger road transport sector, have to be approached with caution, particularly when conducting ex-ante policy impacts projections. The three most important assumptions/ simplifications in this study include: (1) the adoption of a static electricity mix for India, (2) the uncertain real-world driving impacts, particularly for a highly-electrified vehicle, and (3) the absence of a vehicle fleet dynamics. These assumptions, though we believe are justified and, in some cases, necessary to avoid over-complicating the study,

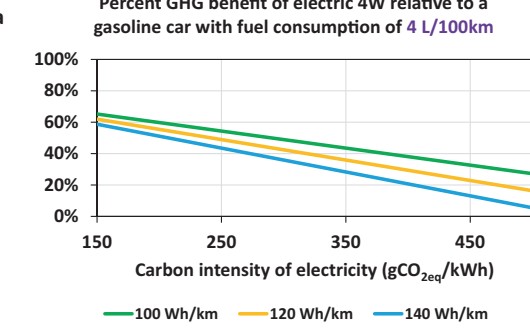

**Percent GHG benefit of electric 4W relative to a gasoline car with fuel consumption of 4 L/100km**

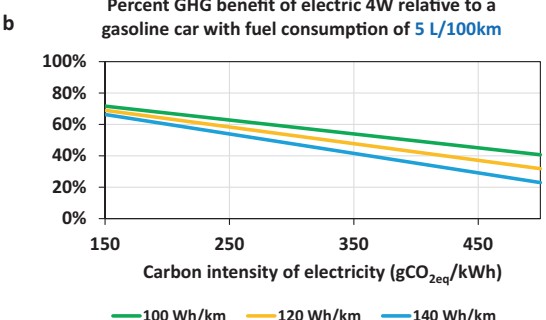

**Percent GHG benefit of electric 4W relative to a gasoline car with fuel consumption of 5 L/100km**

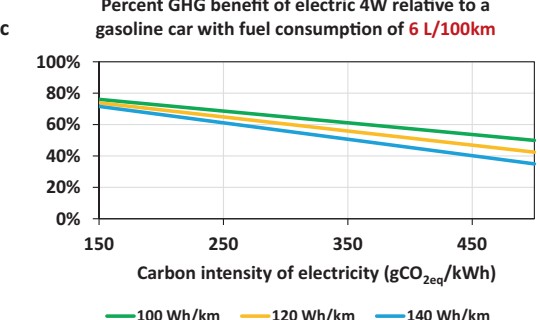

**Percent GHG benefit of electric 4W relative to a gasoline car with fuel consumption of 6 L/100km**

**Fig. 4 Future GHG emissions reduction potential of 4 W BEVs with deeper decarbonisation of the electricity grid.** The life cycle GHG emissions reduction is calculated for a 4 W BEV with different energy consumptions (100, 120, and 140 Wh/km), equipped with a 30 kWh battery, compared to traditional gasoline vehicles with fuel consumptions of (**a**) 4 L/100 km, (**b**) 5 L/100 km, and (**c**) 6 L/100 km.

can have implications on the outcomes, which are briefly discussed below.

First, to estimate the GHG emissions of BEVs in 2018/2019, we assumed a static power generation mix throughout the entire duration of the vehicle lifetime. The power generation profile in India has not improved in the last 20 years. In fact, coal-based power has increased its market share from 69% to 76% between 2000 and 2020 (Supplementary Figure 5). Even in some of the wealthiest cities in India, the per capita electricity consumption is still only about half of the global average[15]. India has the challenge of meeting significant electricity demand growth prospects while ensuring reliable, affordable, and sustainable supply nationwide. The IEA projects that in the short-term, coal-fired electricity generation is set to exceed the pre-pandemic levels in many countries, including India, as its economy and demand for electricity rebounds[64]. Although India has expanded its renewable capacity, about 40 GW of new coal is still under construction[64]. This indicates that India's short-term power sector trend is unlikely to be consistent with a net-zero emissions trajectory. However, for completeness, we present a sensitivity analysis using a dynamic electricity mix for India, with 0.5%, 1.0%, and 1.5% emissions reductions per annum in the supporting document (Supplementary Figure 14). This leads to a slightly lower total GHG emission for BEVs, but more importantly, it further emphasises the importance of deeper and faster decarbonisation of the electric grid. Critically, India must ensure that its post-pandemic recovery is aligned with a 2 °C trajectory.

Second, India's fuel/energy consumption certification is based on the New European Driving Cycle (NEDC). To the best of our knowledge, there isn't an industry-standard correlation between the type-approved vehicle certifications and real-world driving characteristics specific to India. The NEDC is known to significantly underestimate the real-world emissions and energy consumptions of vehicles, with some reporting larger penalties for heavily electrified vehicles[65]. Thus, in this study, we adopted correction factors consistent with the methods in recent literature[6,10,65–69] to adjust the certified energy/fuel consumptions to reflect real-world deteriorations. There is large uncertainty and implication with this assumption (Supplementary Figure 11); however, removing the correction factors (as done in Supplementary Figure 12) ignores the real-world effects that have been well-documented in the literature[70–75]. In a 2021 report, the International Council on Clean Transportation (ICCT) adopted a real-world penalty of 34% for conventional combustion engine vehicles, while the real-world electric range of BEVs would deteriorate by as much as 30% based on empirical evidence[76]. Figure 9 depicts the total life cycle GHG emissions of BEVs,

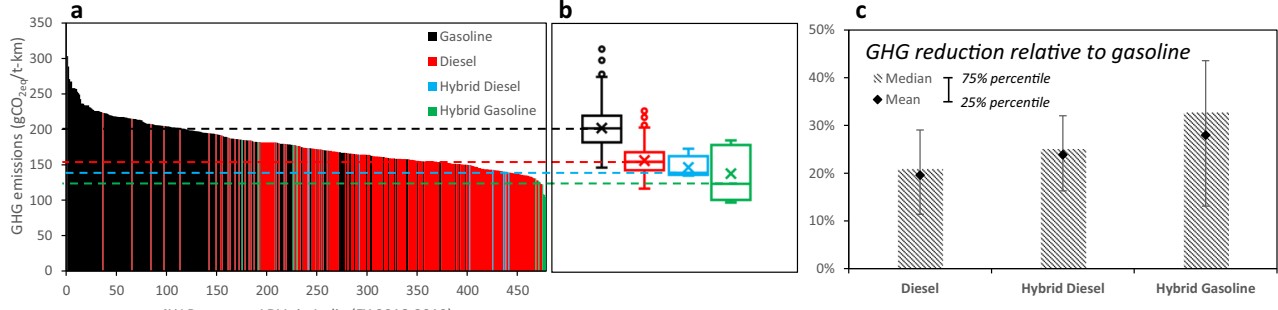

**Fig. 5 GHG emissions intensity of four-wheeled (4 W) passenger road light-duty vehicles (LDVs) in India in 2018/19.** (**a**) Life cycle GHG emissions range for 478 passenger road vehicles in India in gCO$_{2eq}$/ton-km. (**b**) Interquartile GHG emission ranges for gasoline SI, diesel CI, diesel-electric hybrids, and gasoline-electric hybrids. The box-and-whisker plots show the median line, mean marker, 25th / 75th percentile box, min/max whiskers, and potential outliers. (**c**) GHG emission reduction relative to gasoline SI vehicles.

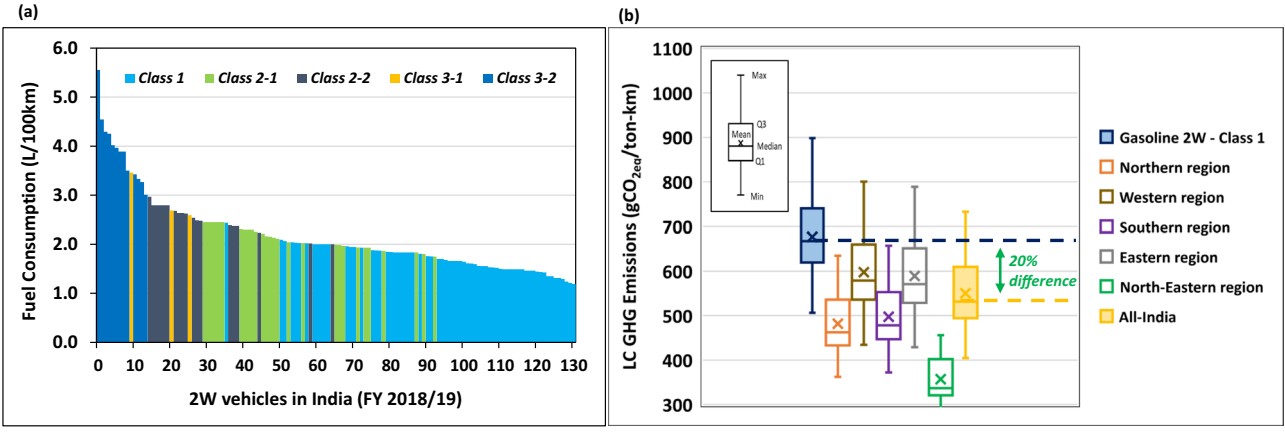

**Fig. 6 Passenger 2-wheelers (2 W) in India (2018/19).** (**a**) Fuel consumptions of 131 conventional gasoline 2-wheelers by class-type available in India in the financial year (FY) 2018/19, in liters/100 km [38]. (**b**) Life Cycle (LC) GHG emissions of electric 2-wheelers charged using the power generation mix in different regions in India and contrasted against 68 Class 1 gasoline-based 2 W models available in 2018/19.

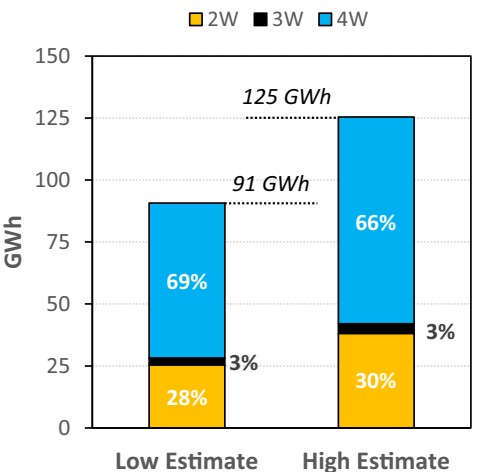

**Fig. 7 Demand for batteries by 2030.** Under the EV30@30 initiative, India would require between 91 and 125 GWh of batteries to electrify its two-wheelers (2 W), three-wheelers (3 W), and four-wheelers (4 W). High estimate and low estimate are based on the variations in battery sizes deployed in 2 W (2kWh & 3 kWh), 3 W (4.7kWh & 6.6 kWh) and 4 W (30 kWh & 40kWh)[43,103]. The percentages refer to the share of battery requirements for 2 W, 3 W, and 4 W. Breakdown of sales forecast by 2030 is provided in Supplementary Figure 7 of the supporting document based on data from Council on Energy, Environment, and Water (CEEW) India[43].

gasoline, and diesel vehicles using the ICCT's real-world correction factors. While this slightly changes individual powertrains' relative GHG reduction merits, it doesn't alter the overall findings.

Third, this analysis quantifies the life cycle GHG emissions of energy and powertrain types on an individual vehicle basis by aggregating the emissions throughout the vehicle lifetime. It is not designed to assess the mitigation potentials on a fleet level. Ultimately, the effectiveness of the technology as a climate change mitigation solution is bound to the overall fleet dynamic, which is often limited by the technology adoption rate and vehicle stock turnover. The former would, in turn, depend on many factors, including consumer acceptability, availability of vehicle models, and infrastructure readiness. On the other hand, discussions are currently ongoing in India to introduce a voluntary vehicle scrappage policy to phase out older vehicles and modernise the fleet. While technology can have a lifecycle GHG reduction potential of 20–30% on an individual vehicle basis, the actual

emissions reduced from the overall fleet critically depend on deployment speed.

## Discussion

There is growing policy interest in BEVs in India—a highly populated subcontinent with an intense appetite for personal mobility. However, with the existing electricity mix, at best, 4 W BEVs offer less than 5% GHG reduction compared to the sales-weighted average vehicle sold in India in 2018/2019 (comprising 65% gasoline and 35% diesel) (Supplementary Figure 22). By 2030, the GHG reduction potential of a 4 W BEV depends critically on how quickly coal can be phased-out from the power generation mix. To offer at least a 20% reduction in overall GHGs, the electricity carbon intensity has to be less than 500 $gCO_{2eq}$/kWh, or well below 350 $gCO_{2eq}$/kWh to attain more than 50% GHG reduction (Supplementary Figure 22). This is 46% and 62% below the current electricity carbon intensity level, implying a near-term and rapid phasing-out of coal from the power sector, which is an unlikely timeline for the EV30@30 ambition. Given India's complicated relationship with the coal industry, even the phasing-down of its use is a large undertaking, as seen in the recently concluded 26th Conference of Parties (COP26) at Glasgow[77].

To improve the prospect for BEVs, India must integrate its mobility transition plan into a broader framework to ensure that its implementation leads to a deep reduction in overall GHG emissions. An organised mobility transition plan supported by phased electrification focused on 2-wheelers, representing 80% of India's passenger road vehicles, is needed. In coordination with the domestic power sector and considering the competing battery supply chains globally, partial electrifications of 4-wheelers through varying degrees of electrified-hybrid solutions also offer credible reductions in transport-related GHG emissions.

However, electrification of road transport can reduce criterion air pollutants, specifically $NO_x$[15] and $PM_{2.5}$[78], which is important for improving ambient air quality, particularly in densely populated cities. On the other hand, without any pollution control measures in place, heavy reliance on coal power plants to generate electricity for a BEV can result in up to 18 times higher emissions of $SO_2$ than a conventional car[15]. This highlights the importance of understanding the trade-offs in policy decisions and the need for a systemic approach to guide India's overall sustainable transition in the transport sector.

A sustainable transport policy has to incentivise the right technology mix at the right time and in the right location. This

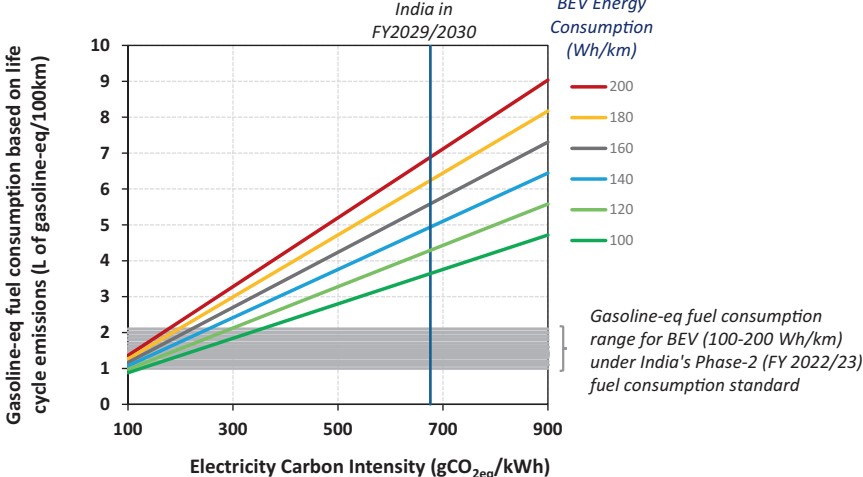

**Fig. 8 GHG loophole created by regulation that is based on tailpipe emission only.** Relationship between the carbon intensity of India's electricity and the gasoline-equivalent fuel consumption of a battery-electric vehicle (BEV) with different efficiencies based on its total life cycle emission. India's Phase-2 standard assigns a gasoline-equivalent fuel consumption of 1–2 L/100 km for BEVs with electricity consumption between 100 and 200 Wh/km regardless of the source of electricity. (Curb weight of BEV = 1237 kg).

implies a need for a bespoke policy mix, specific to a local context and bound by the realities of an interconnected global market – rather than uncritically borrowing wholesale policy features from elsewhere. This is in line with the findings by Wolinetz & Axsen, who demonstrated quantitatively that different policy approaches might better suit a particular regional and technical context[79].

Technology-centric policy on its own is not adequate to meet the sectoral mitigation targets[5], and the 'avoid-shift-improve' framework offers the best chance for achieving sustainable mobility[80,81]. Effective behaviour-centric policies are needed to attenuate demand for road vehicle travel[4] through (i) support for active mobility (cycling and walking), (ii) enhancement of public transit, (iii) optimal design of the built environment, and (iv) reducing the need for travel (telecommute and telework).

There is a propensity to seek a technological fix to obviate the need for changing behaviours whenever a problem becomes formidable[82,83]. But, betting on a single technology implies that it will consistently be the most effective solution across time, sectors, and regions. These broad generalisations are largely indefensible given the preponderance of evidence pointing to the need for a more comprehensive approach[4,5,84,85]. Thus, we need a mosaic of technologies for different transport sectors and geographical regions, including electrified powertrains (counting full BEV), highly efficient combustion engine-powered vehicles, and low-carbon fuels such as biofuels, hydrogen, and $CO_2$-derived fuels, including Methanol and DME. However, these too can only thrive in a sustainable and low-carbon ecosystem, including the accessibility to affordable and clean power, the adoption of improved land-use and sustainable agricultural practices, incentives for private investments in low-carbon technologies, such as carbon capture and storage infrastructures, and the creation of a materials (including battery) recycling and repurposing ecosystem towards achieving a circular carbon economy. For this to happen, policies and regulations must be aligned to reduce the climate change impacts of the overall energy-technology system. Miscounting carbon, particularly in emission regulation, is a serious error that is fixable with adopting a more comprehensive accounting policy framework.

## Methods

**Life cycle assessment framework**. We developed a life cycle assessment (LCA) model for over 600 commercially available passenger-road, light-duty vehicles in India, comprising 2-wheelers and 4-wheelers, which jointly account for 90% of total vehicle sales. We focused on the 2018/19 financial year (April–March), just

before COVID-19, which caused a major disruption globally. The LCA model quantifies the emissions of $CO_2$, $CH_4$, and $N_2O$ from cradle-to-grave, including energy productions, vehicle use-phase, vehicle productions, and vehicle end-of-life recycling, as per Eq. 1.

$$LCE = \frac{E_{ttw} + E_{ep} + E_{vp} - E_{eol}}{CM} \qquad (1)$$

where, $LCE$ = Life cycle emissions in $gCO_{2eq}$/ton-km, $E_{ttw}$ = vehicle tailpipe emissions (from tank-to-wheel) in $gCO_{2eq}$/km, which is zero for BEV, $E_{ep}$ = emissions due to energy production, whether electricity generation or gasoline/diesel fuels production in $gCO_{2eq}$/km, $E_{vp}$ = emissions due to vehicle manufacturing in $gCO_{2eq}$/km, which includes battery production for BEVs, and amortised throughout the total vehicle lifetime kilometers, $E_{eol}$ = net emission due to vehicle end-of-life, which includes the energy use and material recycling, in $gCO_{2eq}$/km, $CM$ = total vehicle curb weight in tonnes.

Emissions due to energy productions ($E_{ep}$) are calculated as per Eq. 2.

$$E_{ep} = CI_e \times EC_v \qquad (2)$$

where, $CI_e$ = carbon intensity of the energy used in $gCO_{2eq}$/MJ for gasoline and diesel fuels, and $gCO_{2eq}$/kWh for electricity. $EC_v$ = vehicle energy consumption measured in MJ/km for gasoline and diesel vehicles, and kWh/km for BEVs.

Detailed assumptions, data sources, and input values used in Eqs. 1 and 2 are provided below.

**Electricity generation ($CI_e$)**. Life cycle GHG emissions intensity, in $gCO_{2eq}$/kWh, was quantified for the power sector in India for the financial year 2018/19 (additional plots provided as Supplementary Figures 3, 4, and 5 in the supporting information). We derived this using the official fuel mix and power generation data from India's Central Electric Authority (CEA), an organisation under the Ministry of Power[21]. The CEA provided data for India and was broken down into five regions: northern, western, southern, eastern, and north-eastern, and hourly fluctuations for a typical day in winter (28 December 2018) and summer (22 June 2018). We assumed a static electricity mix for the duration of the vehicle's lifetime. India-average electricity transmissions and distribution (T&D) losses were obtained from Surana & Jordaan[86] and applied to all the regions.

Future fuels and electricity generations were based on CEA's projection of the optimal mix in 2029/30[87]. India aims to install 175 GW of renewable electricity by 2022, in which 100 GW is the target for solar PV alone[88]. Under the Paris Agreement, it has committed to achieving 40% electric power from nonfossil-based capacities by 2030[88]. The CEA had assessed the optimal generation mix for meeting India's electricity demand in 2029/30 (Supplementary Figure 3), considering feasible technology options, fuel availability, capital and operational costs, climate change commitments, and various constraints, including the intermittency of renewables[87]. India's overall electricity consumption is projected to grow by 82% to 2517 TWh during the period. The CEA expects coal power plants to supply 54% of the total electricity needs[87], down from 74% a decade earlier[22]. By 2029/30, solar and wind power installed capacities are expected to increase to 51% of the total installed capacity[87] from less than 20% in 2018[89], and it is expected to be equipped with 27 GW of battery energy storage system[87]. The average capacity utilisation factors are projected to increase to 25% for wind and

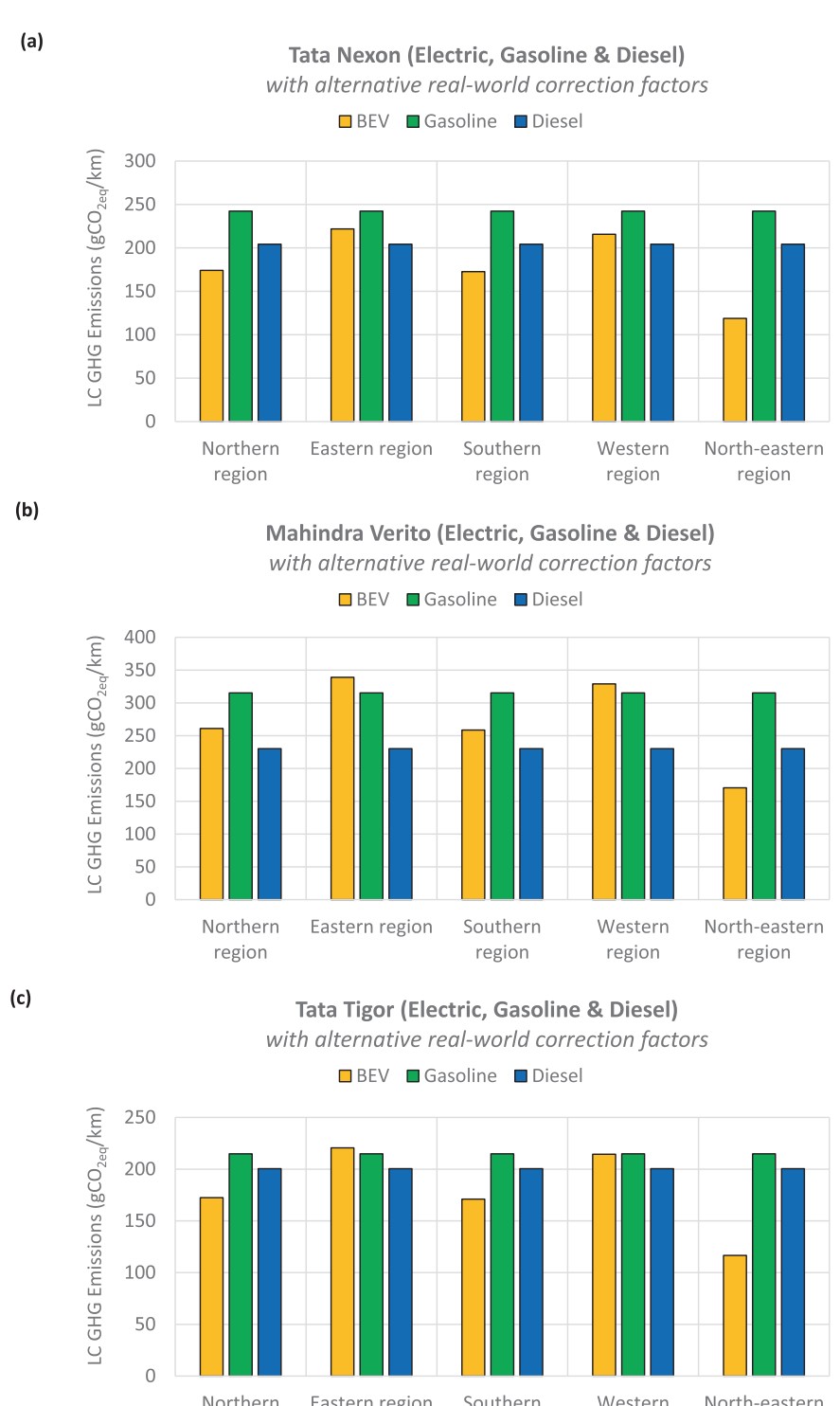

**Fig. 9 The effects of adopting an alternative set of real-world correction factors on the life cycle (LC) GHG emissions of electric, gasoline, and diesel vehicles.** Compared to the NEDC, the electric range of a battery-electric vehicle (BEV) is reduced by 30%, while the fuel consumption of combustion engine vehicles increased by 34% [76]. A total vehicle lifetime distance of 200,000 km was assumed for (**a**) Tata Nexon, (**b**) Mahindra Verito, and (**c**) Tata Tigor.

22% for solar[87,89]. Still, it is estimated to supply only 31% of India's overall electricity generation mix in 2029/30[87].

The average electricity mix in India resulted in carbon intensity ($CI_e$) of 927 and 676 gCO$_{2eq}$/kWh for 2018/19 and 2029/2030.

**Fuel productions ($CI_e$).** The well-to-tank carbon intensities of traditional gasoline and diesel fuels include crude oil extraction and refining. A specific crude oil mix and refinery emissions for India were derived from Masnadi et al[90]. and Jing

et al.[91], respectively. This resulted in well-to-tank carbon intensity ($CI_e$) of 17.3 and 16.4 gCO$_{2eq}$/MJ for gasoline and diesel, respectively. The combustion-generated emissions for the two fuels were based on the efficiency and vehicle tailpipe emissions given below.

**Vehicle efficiency & tailpipe emissions ($EC_v$ & $E_{ttw}$).** India's corporate average fuel consumption standards are based on a modified NEDC drive cycle. This study utilised vehicle energy consumptions and exhaust emissions data, certified by the

Automotive Research Association of India (ARAI). It is collated from various sources, primarily from the Society of Indian Automobile Manufacturers (SIAM)[38] for vehicles commercially available during the 2018/19 financial year (raw data provided in Supplementary Tables 1 and 2, and Supplementary Figures 1 and 2 in the supporting information). Type approvals for vehicle energy consumption and resulting tailpipe emissions, particularly under the NEDC drive cycle, are known to underestimate the real-world efficiency and emissions by an average margin of 10-40% based on empirical findings in the USA, Europe, and China[70–75]. Consistent with methodologies reported in the literature[6,10,65–69], we adjusted the type-approved energy consumptions to reflect real-world deteriorations by a 20% penalty factor for ICEV and HEV and a 40% penalty factor for BEV, based on the analyses by Argonne National Laboratory. We further applied temperature adjustment factors to the real-world energy consumption rates to incorporate ambient temperatures on vehicle operation in different regions in India (Supplementary Figure 10). The adjustment factors estimated the impacts of ambient temperatures on battery efficiency and the use of cabin air-conditioning[6,12,92–94]. In this study, we adopted the approach used by Wu et al.[6], in which the estimated India-average energy consumption penalties were 4%, 5%, and 6% for ICE, HEV, and BEV, respectively (Supplementary Figure 10), based on the official monthly mean temperatures recorded in India[95]. Sensitivity analyses on the impact of ambient temperatures on the overall life cycle emissions are presented in the supporting document (Supplementary Figures 12 and 13).

**Vehicle production and end-of-life recycling ($E_{vp}$ & $E_{eol}$).** Emissions due to 4 W vehicle production and recycling were mainly extracted from Sphera's LCA software and databases (GaBi version 9.2 with 2020 LCI Databases)[96] (Supplementary Figure 6). As India does not yet have a robust vehicle scrappage policy, this study assumed a total vehicle lifetime travel distance of 200,000 km (i.e., 12,500 annual kilometers[97,98] over 16 years), roughly in-line with other studies[76]. Data associated with battery manufacturing and recycling in GaBi were replaced with a recent LCA report in the literature. The report used primary data from two leading Chinese corporations for each life cycle stage: Lithium battery manufacturing, cathode materials production, and battery recycling[99]. This resulted in a battery manufacturing carbon intensity of 124.5 kgCO$_{2eq}$/kWh, and when battery end-of-life recycling was included, the net carbon intensity was reduced to 93.6 kgCO$_{2eq}$/kWh. We estimated that for Indian BEVs with curb weights between 1200 and 1600 kg and equipped with batteries in the range of 20-30kWh, the resulting vehicle manufacturing emissions ranged between 8 and 10 tons of CO$_{2eq}$. Meanwhile, the end-of-life recycling credit was between 3 and 4 tons of CO$_{2eq}$[96]. Thus, the overall emissions for the vehicle life cycle were 5–6 ton CO$_{2eq}$.

There are limited data available in the literature for emissions associated with the production and recycling of 2-wheelers. We estimated the production emissions using a linear-by-mass relation derived from the 4-wheelers and resized several components accordingly (e.g., number of tires, fuel tank size, and battery capacity). For a class-1 2-wheeler with a curb weight of 109 kg, the net production and recycling emissions were estimated to be about 1.1 ton CO$_{2eq}$, resulting in an overall life cycle GHG emissions of about 73.8 gCO$_{2eq}$/km. For comparison (Supplementary Figure 16), Cox & Mutel reported that the total production emissions for European gasoline 2 W, with curb weights of 90-198 kg, was about 0.7–2.6 ton CO$_{2eq}$, resulting in overall life cycle emissions of 81–176 gCO$_{2eq}$/km[100]. For electric 2 W, with battery sizes in the range of 1.0–3.3 kWh and curb weights between 68 and 118 kg, we estimated the net manufacturing and recycling emissions to range between 1.0 and 1.2 ton CO$_{2eq}$ (Supplementary Fig. 16 provides a comparison against the values derived by Cox & Muttel[100]. Based on the literature, we assumed a lifetime mileage of 80,000 km for the 2 W[100,101]. Sensitivity analyses were conducted to assess the impacts of adopting different assumptions on the life cycle GHG emissions of the 2 W (Supplementary Figs 17 and 18).

## Data availability

The data supporting this study's findings are publicly available, and the sources are documented in the methods section and presented in the Supporting Information. Supplementary Tables 1, 2, and Supplementary Figures 1-6 in the supporting information, and Table 1 in the main manuscript, contain raw data used in the analyses. The excel models, with data sources, are available from the corresponding author upon reasonable request.

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

## Acknowledgements

The authors are solely responsible for the contents of the paper and do not necessarily represent the views of any organizations.

## Author contributions

A.F.N.A.M., A.K.A., G.K. and A.A.A. conceptualized the study. A.F.N.A.M conducted the analysis. A.F.N.A.M. and V.G.Z. developed the methodology and models. A.F.N.A.M., V.G.Z., A.K.A., G.K. and A.A.A. contributed in the writing and editing of the paper.

## Competing interests

The authors declare no competing interests.
