## [Peer Review File · Nature Communications]

Reviewer comments, initial review -

Reviewer #1 (Remarks to the Author):

The abstract needs to be edited to clearly support the conclusions. The paper is really two papers- it is a life cycle assessment of electric four-wheeled (light-duty) vehicles in India, and a life cycle assessment of two-wheeled electric motorscooters in India. The data, methods, and assumptions for the four wheeler analysis and future grid carbon intensities do not support the conclusions. There is some sensitivity analysis done in the SI, but not really discussed in the main text, policy implication, and conclusions. The 2 wheeler analysis is an important contribution and highlights the need to electrify this sector. However, these are two different transportation modes and are not always directly substitutable. All data, code, and models need to be publicly available for a paper like this to ensure replicability.

Line 4: Remove first word The

Line 6: "Electric 2-wheelers can enable a 20% reduction in overall GHG emissions given their small battery size and improved powertrain efficiency" state whether this and other results in the abstract are under current conditions or some future grid condition.

Line 10: charged in the summer

Line 11: "Electrified 4-wheelers, such as gasoline-electric hybrids, can lead to 33% GHG reductions, though they haven't been fully considered as a mitigation option in India". If referring to plug-in hybrid vehicles, say plug-in hybrids rather than gasoline-electric hybrids, which do not plug in. Also make clear in this sentence if this paper has evaluated them.

Line 23: "Electrification of transport is a good example of an approach believed to be a panacea". A reader might take away from this that there are other currently viable fuels and powertrains for decarbonized light-duty vehicle transportation besides electricity, when there are not. LDV electrification needs to be complemented with grid decarbonization, demand reduction, and public transit expansion. But for light-duty vehicle fuels, is there another currently viable decarbonization option? If not please revise this panacea sentence.

Line 25: spell out GHG on first use

Line 26: "Therefore, the power generation mix has to be adequately decarbonized for the benefits of a battery electric vehicle (BEV) to materialize". This is not accurate as written. One could say "adequately decarbonized for battery electric vehicles (BEVs) to enable deep reductions in LDV GHG emissions." Even when charged completely on coal power, BEVs have GHG benefits compared to the average vehicle. As the grid GHG intensity moves about 400 g GHG/kWh and below (similar to natural gas power emissions), the GHG benefits are larger, and then under decarbonized power, the GHG benefits are very large. So the sentence needs to be reworded to emphasize that the GHG benefits materialize immediately but continue to grow as the grid is decarbonized. There are several studies that could be cited to support this point. It also is important to note that the benefits referred to here are GHG benefits, rather than the air pollutant benefits.

Line 28: "Diversity in power generation profiles and differences in fleet characteristics [5] implies that the BEVs' potential for climate change mitigation cannot be generalized worldwide". Again, this is under current conditions not future grid conditions, even over the 15 year lifetime of a vehicle purchased today. And if a BEV is replacing the current average fleet vehicle, even when charged on coal it would reduce GHGs a little.

Line 29: "implies that the BEVs' potential for climate change mitigation cannot be generalized worldwide" same issue as above. This sentence needs a reference or needs to be reworded/removed.

Line 39: specify GHG or air pollutant emissions

Line 41: is this total only privately-owned LDVs?

Line 49: would be good to allude to what types of policy interventions are necessary

Line 54: production

Line 62: "We find that the electrification of 2-wheelers presents an enormous opportunity to reduce GHG emissions: an electric 2W is substantially more energy-efficient than a traditional gasoline 2W, and it is equipped with a smaller battery pack therefore, it carries a much lower GHG burden compared to an electric 4W. However, the climate change mitigation potential of 4W BEVs is critically dependent on how they are charged." The research question is unclear here. What is the climate change mitigation potential of a 2W compared to- an electric 2W compared to a petrol-powered 2W? Or are 2Ws compared to 4Ws? These are different vehicles and while uses overlap, they cannot be directly compared. 2Ws are much smaller and lighter and have less passenger and cargo space. So is the question here, should India electrify 2W to replace petrol vehicles? Or is the question should India encourage electric 2Ws over petrol and electric powered 4W? The comparison needs to be clear. Also, the conclusion that the level of mitigation potential for BEVs depends on how they are charged is well established in the literature and needs supporting citations here.

Line 72: "Although climate change is a global challenge, there is no universal, one-size-fits-all technology solution and policy solution." This is largely true, except for the LDV sector. Behavioral modifications are surely needed but these alone will not decarbonize the LDV sector. Is the answer different for India?

Figure 1: the figure shows the benefits of vehicles as positive and negative benefits as increased emissions. Along the lines of the paper's framing, the authors could consider having Figure 1 be "emissions reduction potential" rather than benefits, with negative values as a reduction in GHGs. Figure 3 and 4 use alternating signs to denote GHG benefits.

Figure 1: needs to state if the emissions factors of electricity are static over the vehicle's life. If so, is this a supported assumption for India? Will the grid GHG intensity fall at all over the next 15 years? Has it fallen in the last 15?

Figure 2: This figure shows the life cycle emissions of BEVs for summer and winter charging depending on time of day for the 2018 grid. The figure is confusing, because there are two components to LCA emission- the direct emissions and the upstream emissions. The upstream emissions are static and allocated per km of travel over the entire lifetime of the vehicle. The direct emissions for that specific trip will vary depending on how the vehicle is charged, and the total life cycle emissions of the vehicle per km of its life will depend on the charging profiles over its entire life. These charging profile assumptions are not stated across vehicles and sensitivity analysis is needed. This figure implicitly is showing how the grid GHG intensity varies by time of day and seasonally. But what is not shown here and made clear is when the vehicle starts or stops its charging cycle in order to result in these life cycle emissions, or the speed/power level of the assumed charging. The grid intensity changes over the course of a 24-hour period and the life cycle emissions for that driving charged with that single day's hourly grid intensity (with appropriate description of the time charging starts and stops in the caption), but the life cycle emissions over the vehicle life requires a different figure showing this info and the assumptions on how the grid changes. The figure needs to be revised to support the conclusions.

Line 112: "Correspondingly, it allows BEVs in 2030 to reduce their life cycle emissions by 23-25% (Figure S13) relative to a BEV in the year 2018/19". This is true under these assumptions for a BEV purchased in 2019 and also driven in 2030. But a future BEV purchased in 2030 will have a cleaner grid and also be more efficient to drive (lower Wh/km), due to technology improvement, competition, and potentially regulation. Need to make this clear in the text and calculations.

Line 115: "Phase 2 of the fuel consumption standard in India, which comes into effect from the

year 2022/23 onwards [30], stipulates that 4W passenger vehicles with an average curb weight of 1237 kg should not exceed 4.95L/100km. Its overall GHG emissions would be comparable to a BEV with an energy consumption of 142 Wh/km (Figure 3) for a gasoline spark-ignition vehicle." Two of the three existing 4Ws in India listed by the authors in Table 1 have energy consumption values of well below 143 Wh/km (100 and 101). Energy efficiency of BEVs is also potentially improved in the future as per my previous comment. In addition "it's overall GHG emissions" would not be comparable at this value- it's GHG emissions for that year for that specific grid carbon intensity, which is likely to continue to fall over the many years of the vehicle's life.

Figure 3: Does this model how the grid changes over the course of the vehicle's life? If not, the Figure does not support the conclusions. What are the assumptions about what time of day the vehicle is charged? Are these life cycle emissions? Need clarification in the caption.

Figure 3: at editor's discretion: change kerb to curb for consistency throughout

Figure 4: This figure shows GHG reduction potentials for all BEVs for any grid cleaner than 500 g/kWh. So doesn't it show that BEVs have a mitigation potential and contradict the message of the paper?

Line 138: "gasoline-electric and diesel-electric hybrids" it looks like these are plug-in hybrid electric vehicles (PHEVs) rather than hybrid-electric vehicles (HEVs) which do not plug in and are mentioned later in the sentence, and need to be denoted as PHEVs, and the all-electric range stated. However in the next sentence, it says that no additional infrastructure is required for PHEVs, which is not the case.

Line 144: This conclusion is only for an electric grid that does not get cleaner

Line 186: please clarify how a 2W removes range anxiety or remove that part of the sentence

Line 191: vehicle sales

Line 190: The entire critical materials section is very well done. Nice job.

Line 232: define NEDC

Line 249: I don't know if it's a mistake, rather than an undercounting- there could be innovation, manufacturing, or air pollutant avoidance reasons for favoring EVs in tailpipe regulations. There are several papers about how existing fuel economy regulations in various countries undercount EVs in their fuel economy regulations. Those should be cited here.

Line 268: "4W BEVs do not offer many prospects for reducing transport GHG in India within the timeframe of this study" this is not supported by the authors' calculations and results.

Line 294: finish sentence

Line 295: "highly efficient combustion engine-powered vehicles, and low-carbon fuels such as biofuels, hydrogen, and CO₂-derived fuels, including Methanol and DME". All of these fuels currently, even when CCS is used (which is not yet in use), generate some or a substantial amount of CO₂ and are not a zero carbon solution.

Methods: The authors' values used for a Tata Nexon and 927 g CO₂/kWh in the current Indian grid generate direct CO₂ values of 93 g/km for the EV, 114 g/km for the gasoline vehicle, and 118 g/km for the diesel vehicle. Even with this very high electricity carbon intensity that will come down, the EV is the best choice (from a direct perspective, and upstream values will add some but unlikely flip this answer as both EVs and petrol have upstream impacts). The paper is written and the conclusions stated that 4Ws are worse or negligible.

Line 370: the authors used a 40% degradation of real world fuel economy for BEVs, referring to Argonne National Laboratory and citing several older reports and conference papers. This 40%

number is inconsistent with most BEV studies. Are these citations for PHEVs? This assumption alone has the ability to change the answer and sensitivity analysis is needed.

Line 385: the assumption of 124.5 kg/kWh of BEV battery manufactured is at the upper end of the distribution of values in the literature, and about double the values used by Argonne National Laboratory. What happens if lower values are used/achieved?

Line 390: please provide a reference for the recycling credit assumption.

Line 410: "The excel models, with data sources, are available from the corresponding author upon reasonable request." For a paper like this, in order to be published all of these models need to be publicly available online with a DOI.

Reviewer #2 (Remarks to the Author):

I have gone through the revisions made by the authors and satisfied with the revisions made to the paper

Reviewer #3 (Remarks to the Author):

The revised manuscript provides clarity and the reviewer has enjoyed reading the article. The data provided and methodology is clearly stated such that the work can be replicated by others. The authors have provided proper justifications to the queries raised in the previous review. However, the reviewer requires justification for following

1. The Figure.2 describes the hourly fluctuations of GHG emission for charging a vehicle in winter and summer. Please provide the details of charging considered, i.e., whether the battery is charged by a level-1, level-2 or level-3 charger. For example, if a 10kWh battery is charged by a level-1 charger of 1kW, the energy consumed from the grid is around 1kWh over the period of 10 hour. This would reduce the GHG emission during that window of the charging when compared to level-3 charging that utilises total 10kWh of energy in an hour, thereby increasing the emissions.
2. The author is suggested to also consider the emissions during manufacturing of EV and the emissions during manufacturing of replacements components in EV required due to faults over its lifetime.
3. The authors have mentioned 200,000 kms as life time for Four wheeler but no proper referencing of the source is provided. Justify how authors came to conclusion of 200,000 kms as life of EV.
4. The author mentions "The life cycle emissions of BEVs oscillate over a wide range depending on when the charging takes place." Provide justification.

Response to reviewers

Reviewer #1 (Remarks to the Author):

The abstract needs to be edited to clearly support the conclusions. The paper is really two papers- it is a life cycle assessment of electric four-wheeled (light-duty) vehicles in India, and a life cycle assessment of two-wheeled electric motorscooters in India. The data, methods, and assumptions for the four wheeler analysis and future grid carbon intensities do not support the conclusions. There is some sensitivity analysis done in the SI, but not really discussed in the main text, policy implication, and conclusions. The 2 wheeler analysis is an important contribution and highlights the need to electrify this sector. However, these are two different transportation modes and are not always directly substitutable. All data, code, and models need to be publicly available for a paper like this to ensure replicability.

We wish to thank the reviewer for the detailed review of our manuscript, and for offering many constructive feedbacks to improve the quality of our analysis and write-up. We have made substantial revisions to the manuscript, which include the following:

(a) to carefully revise the language throughout the manuscript, particularly in the conclusions, to ensure that it is more in-line with the findings of our analyses. This also involves providing further clarifications throughout the manuscript as per reviewer's suggestion;

(b) replaced Figure 2 on the effects of battery re-charging time and duration based on the temporal fluctuations of the grid generation mix in the summer and winter. This is a very good point made by the reviewer; and

(c) we added a new section, titled "Reflecting on the limits of generalizations", towards the end of the manuscript (just before the Methods section) to discuss the implications of our modelling choices and assumptions on the overall findings. This include sensitivity analyses on 2 of the most important points raised by the reviewer (i.e., effects of static vs dynamic grid mix, and effects of adopting a different set of real-world correction factors for BEVs and traditional vehicles).

Furthermore, we have also made the revisions as per the reviewer's specific comments/suggestions below.

Line 4: Remove first word The – Revised as per reviewer suggestion.

Line 6: "Electric 2-wheelers can enable a 20% reduction in overall GHG emissions given their small battery size and improved powertrain efficiency" state whether this and other results in the abstract are under current conditions or some future grid condition. Clarified as per reviewer's suggestion, while still adhering to the journal's 150words limit for abstract:

"Currently, electric 2-wheelers can enable about 20% reduction in overall GHG emissions given their small battery size and improved powertrain efficiency."

Line 10: charged in the summer - Revised as per reviewer suggestion.

Line 11: “Electrified 4-wheelers, such as gasoline-electric hybrids, can lead to 33% GHG reductions, though they haven’t been fully considered as a mitigation option in India”. If referring to plug-in hybrid vehicles, say plug-in hybrids rather than gasoline-electric hybrids, which do not plug in. Also make clear in this sentence if this paper has evaluated them. *We did not assess plug-in hybrids (PHEV) given that they are not commercially available in India. Thus, we have focused on HEVs. We have made this clear in the manuscript. The journal has a strict 150 words limit for the abstract and therefore we are constrained by how much details we can share in the abstract. We are already very close to the maximum word limit. However, we revised this sentence as per the following, to indicate that we refer to a self-charging hybrid, and not a plug-in hybrid:*

“Self-charging gasoline-electric hybrids, can lead to 33% GHG reductions, though they haven’t been fully considered as a mitigation option in India”

Line 23: “Electrification of transport is a good example of an approach believed to be a panacea”. A reader might take away from this that there are other currently viable fuels and powertrains for decarbonized light-duty vehicle transportation besides electricity, when there are not. LDV electrification needs to be complemented with grid decarbonization, demand reduction, and public transit expansion. But for light-duty vehicle fuels, is there another currently viable decarbonization option? If not please revise this panacea sentence.

We have removed the word panacea in this sentence and revised it accordingly:

“Electrification of transport is a good example of a technological-fix that has been the narrow focus of many government policies and mandates at the expense of a more comprehensive approach that is much needed [4, 5].”

Line 25: spell out GHG on first use - *Revised as per reviewer suggestion.*

Line 26: “Therefore, the power generation mix has to be adequately decarbonized for the benefits of a battery electric vehicle (BEV) to materialize”. This is not accurate as written. One could say “adequately decarbonized for battery electric vehicles (BEVs) to enable deep reductions in LDV GHG emissions.” Even when charged completely on coal power, BEVs have GHG benefits compared to the average vehicle. As the grid GHG intensity moves about 400 g GHG/kWh and below (similar to natural gas power emissions), the GHG benefits are larger, and then under decarbonized power, the GHG benefits are very large. So the sentence needs to be reworded to emphasize that the GHG benefits materialize immediately but continue to grow as the grid is decarbonized. There are several studies that could be cited to support this point. It also is important to note that the benefits referred to here are GHG benefits, rather than the air pollutant benefits.

Thank you for this comment. We have amended the sentence as per reviewer’s suggestion. See below:

“Therefore, the power generation mix must be adequately decarbonized to enable deep reductions in GHG emissions from the passenger road LDV sector.”

Line 28: “Diversity in power generation profiles and differences in fleet characteristics [5] implies that

the BEVs' potential for climate change mitigation cannot be generalized worldwide". Again, this is under current conditions not future grid conditions, even over the 15 year lifetime of a vehicle purchased today. And if a BEV is replacing the current average fleet vehicle, even when charged on coal it would reduce GHGs a little. An important consideration is the segment of the vehicle fleet that will actually be displaced by the BEVs, as this will have significant impacts on the GHG reduction potentials of the BEV. The GHG benefit of the BEV diminishes when contrasted against the more fuel-efficient vehicles, for example a diesel engine (i.e. about 15-20% more efficient than a traditional gasoline vehicle), or a HEV (i.e. typically 20-25% more efficient than an average gasoline car). Therefore, the magnitude of the GHG reduction (i.e. GHG reduction potentials) will be context specific. We agree with the reviewer that even coal-powered BEVs can reduce GHGs slightly when contrasted against the average vehicle fleet (i.e. composed of traditional gasoline engines, with a variety of sizes including the large and fuel-intensive SUVs). We have amended the text accordingly and added 3 references, as per below:

"Therefore, the power generation mix must be adequately decarbonized to enable deep reductions in GHG emissions from the passenger road LDV sector. In some geographical regions with high clean and renewable power penetration, battery-electric vehicles (BEVs) offer a significant reduction in transport GHGs [6, 7, 8, 9, 10]. Diversity in power generation profiles and differences in fleet characteristics [5] implies that the BEVs' GHG reduction potentials cannot be generalized worldwide [11, 12, 6]."

Line 29: "implies that the BEVs' potential for climate change mitigation cannot be generalized worldwide" same issue as above. This sentence needs a reference or needs to be reworded/removed. Similar to the response above, we have moderated the text slightly, and added 3 references.

"Therefore, the power generation mix has to be adequately decarbonized to enable deep-reductions in GHG emissions from the passenger road LDV sector. In some geographical regions with high penetration of clean and renewable power, battery-electric vehicles (BEVs) offer significant reduction in transport GHGs [6, 7, 8, 9, 10]. Diversity in power generation profiles and differences in fleet characteristics [5] implies that the BEVs' GHG reduction potentials cannot be generalized worldwide [11, 12, 6]."

Line 39: specify GHG or air pollutant emissions – We have added clarification that this refers to GHGs.

Line 41: is this total only privately-owned LDVs? This includes 2W, 3W, and 4W road vehicles. We have added this clarification in the amended text. Some of the 3W vehicles are also used in a shared mobility and as public transport. However, we do not have the split between privately-owned and commercially-owned vehicles.

Line 49: would be good to allude to what types of policy interventions are necessary. We have amended the text following the reviewer's suggestion, as per below:

"With a population of more than 1.3 billion, accounting for 17.8% of the global population in 2019, a lack of broad policy interventions encompassing the 'avoid-shift-improve' framework can significantly increase GHG emissions from the Indian transport sector."

Line 54: production - Revised as per reviewer suggestion.

Line 62: “We find that the electrification of 2-wheelers presents an enormous opportunity to reduce GHG emissions: an electric 2W is substantially more energy-efficient than a traditional gasoline 2W, and it is equipped with a smaller battery pack therefore, it carries a much lower GHG burden compared to an electric 4W. However, the climate change mitigation potential of 4W BEVs is critically dependent on how they are charged.” The research question is unclear here. What is the climate change mitigation potential of a 2W compared to- an electric 2W compared to a petrol-powered 2W? Or are 2Ws compared to 4Ws? These are different vehicles and while uses overlap, they cannot be directly compared. 2Ws are much smaller and lighter and have less passenger and cargo space. So is the question here, should India electrify 2W to replace petrol vehicles? Or is the question should India encourage electric 2Ws over petrol and electric powered 4W? The comparison needs to be clear.

We agree with the reviewer that the way the sentence was constructed can be confusing, especially when the earlier version of the manuscript made a direct comparison between the battery size of 2W and 4W. This study is mainly assessing the GHG effects of road transport electrification – i.e., what is the GHG consequences of electrifying 2W and 4W by contrasting them against their respective traditional gasoline counterparts. It is not a comparison between 2W and 4W. We have removed the comparison between 2W and 4W to avoid ambiguity, as per revised text below:

“We find that the electrification of 2-wheelers presents a big opportunity to reduce GHG emissions: an electric 2W is substantially more energy-efficient than a traditional gasoline 2W, and it is equipped with a small battery pack therefore, it carries only a minor manufacturing GHG penalty. However, the climate change mitigation potential of 4W BEVs is critically dependent on how they are charged [8, 7, 11, 6]. The time of the day and the year when the BEV is charged, and the region where the charging occurs has a considerable effect on its overall life cycle GHG emissions.”

Also, the conclusion that the level of mitigation potential for BEVs depends on how they are charged is well established in the literature and needs supporting citations here. Agreed. We have added 4 references there (see above).

Line 72: “Although climate change is a global challenge, there is no universal, one-size-fits-all technology solution and policy solution.” This is largely true, except for the LDV sector. Behavioral modifications are surely needed but these alone will not decarbonize the LDV sector. Is the answer different for India? There are several technological options for decarbonizing passenger road transport. Electrification is certainly one leading option that has gained considerable interests worldwide recently. Other technologies include hydrogen fuel cells and the use of advanced low-carbon renewable fuels such as biofuels and synthetic electrofuels (derived from captured CO₂ and renewable electricity). Each technology has its own limitations and merits, and these can vary from region to region. Countries in the Far East, like China and Japan, are fast-tracking their hydrogen economy. Low carbon fuels (e.g. biofuels and electrofuels) will also play a role in some markets – in the short to medium term, the US and the EU have their low carbon fuels mandates as a means to reduce emissions from the LDV sector, particularly for the existing fleet where the vehicle turnover rate in the market might be limiting. Globally, we will require a mix of technologies contributing towards decarbonizing the passenger road LDV, but none of it is sufficient on its own.

As for India, and many other non-Annex 1 countries, their policies have been heavily focused on electrifying their LDV fleet (refer to the NDC under the Paris Agreement). Electrification has an important role to play, but it has to be accompanied by policies to accelerate grid decarbonization. Unfortunately, there is also a lot less emphasis on the behavioral-modification aspects of the avoid-shift-improve framework. The reviewer is right that behavioral-modification policies on its own is insufficient. But likewise, the electrification of LDVs alone will not be enough to meet the sectoral mitigation targets, as demonstrated by Milovanoff et al. in a recent 2020 Nature Climate Change paper (<https://www.nature.com/articles/s41558-020-00921-7>).

Here, we call for India to adopt a broad mix of policy and technology solutions to mitigate emissions from the sector, and to remind readers and policymakers that there is no simple solution or silver-bullet.

Figure 1: the figure shows the benefits of vehicles as positive and negative benefits as increased emissions. Along the lines of the paper's framing, the authors could consider having Figure 1 be "emissions reduction potential" rather than benefits, with negative values as a reduction in GHGs. Figure 3 and 4 use alternating signs to denote GHG benefits. Throughout the manuscript (including Figures 1, 3, and 4), we have used positive values to denote benefit (i.e. GHG reduction), and negative values where the emissions are higher (i.e. a disbenefit). However, to avoid confusion, we have revised the caption accordingly to add more clarity, as per the following for Figure 1:

"GHG emission reduction potential of electric 4-wheelers in India. A positive value denotes the percentage emission reduction while a negative value means that the BEV emits higher GHG than its gasoline/diesel counterparts."

Figure 1: needs to state if the emissions factors of electricity are static over the vehicle's life. If so, is this a supported assumption for India? Will the grid GHG intensity fall at all over the next 15 years? Has it fallen in the last 15?

We thank the reviewer for bringing this up. This is an important point. The reviewer is correct that we have assumed a static electricity mix for India's power generation since it is hard to justify a more optimistic assumption on the rate of grid decarbonization in the short-term, particularly in a post-pandemic India with the expected rebound growth in the economy and the subsequent rise in demand for electricity.

This requires a bit more explanation and investigation. So, we have done two things here:

- 1) As per reviewer's suggestion, we revised the caption for Figure 1 to state that the analysis is based on a static generation mix. See below.

"Figure 1 GHG emission reduction potential of electric 4-wheelers in India. A positive value denotes the percentage emission reduction while a negative value means that the BEV emits higher GHG than its gasoline/diesel counterparts. The life cycle GHG emissions of BEVs in India vary significantly based on individual regions' power generation profiles, with the generation mix assumed to be static throughout the vehicle lifetime. There are larger overall GHG emission benefits for BEVs in the north-eastern region and to a lesser extent in the northern and southern regions. The climate change mitigation potential of a BEV is diminished when contrasted against diesel engines. Figure S8 provides additional details".

2) We have also added a new section in the report, titled “Reflecting on the limits of generalizations” to discuss the implications of our methodological choices and assumptions. Please find below the discussion pertaining to the effects of adopting a static vs dynamic electricity mix. We also conducted a sensitivity analysis and appended a new plot in the supporting document as Figure S14. We thank the reviewer for this good suggestion.

“First, to estimate the GHG emissions of BEVs in 2018/2019, we assumed a static power generation mix throughout the entire duration of the vehicle lifetime. The power generation profile in India has not improved in the last 20 years, in fact coal-based power has increased its market share from 69% to 76% between the years 2000 and 2020 (Figure S5). Even in some of the wealthiest cities in India, the per capita electricity consumption is still only about half of the global average [15]. India has the challenge of meeting significant electricity demand growth prospects while ensuring reliable, affordable, and sustainable supply nationwide. The IEA projects that in the short-term, coal-fired electricity generation is set to exceed the pre-pandemic levels in many countries, including India, as its economy and demand for electricity rebounds [74]. Although India has expanded its renewable capacity, about 40 GW of new coal capacity is still under construction [74]. This indicates that India's short-term power sector trend is unlikely to be consistent with a net-zero emissions trajectory. However, for completeness, we present a sensitivity analysis using a dynamic electricity mix for India, with 0.5%, 1.0%, and 1.5% emissions reductions per annum in the supporting document (Figure S14). This leads to a slightly lower total GHG emission for BEVs, but more importantly, it further emphasizes the importance of deeper and faster decarbonization of the electric grid. Critically, India must ensure that its post-pandemic recovery is aligned with a 2°C trajectory.

Figure S5 Electricity generation profile in India in the last 20 years from the year 2000 to 2020. Data were derived from the IEA [5] and India’s CEA [2].

Figure S14 The effects of adopting a dynamic carbon intensity profile on the life cycle GHG emissions of BEVs in India. (a) Three emissions improvement rates (0.5%, 1%, and 1.5% per year) were assessed relative to a static electricity base case. (b) The net GHG emissions of BEVs over the vehicle life cycle under the different grid decarbonization scenarios.

Figure 2: This figure shows the life cycle emissions of BEVs for summer and winter charging depending on time of day for the 2018 grid. The figure is confusing, because there are two components to LCA emission- the direct emissions and the upstream emissions. The upstream emissions are static and allocated per km of travel over the entire lifetime of the vehicle. The direct emissions for that specific trip will vary depending on how the vehicle is charged, and the total life cycle emissions of the vehicle per km of its life will depend on the charging profiles over its entire life. These charging profile assumptions are not stated across vehicles and sensitivity analysis is needed. This figure implicitly is showing how the grid GHG intensity varies by time of day and seasonally. But what is not shown here and made clear is when the vehicle starts or stops its charging cycle in order to result in these life cycle emissions, or the speed/power level of the assumed charging. The grid intensity changes over the course of a 24-hour period and the life cycle emissions for that driving charged with that single day's hourly grid intensity (with appropriate description of the time charging starts and stops in the caption), but the life cycle emissions over the vehicle life requires a different figure showing this info and the assumptions on how the grid changes. The figure needs to be revised to support the conclusions.

We completely agree with the reviewer. We have recalculated the life cycle emissions and replaced Figure 2 with a new plot, which now takes into account of the BEV time-of-charge and the charging duration. The latter is estimated from the battery size and the power of the charger. We thank the reviewer for this good observation and feedback. See below for the updated plot.

Figure 2 Effects of battery recharging time on the life cycle GHG emissions of BEVs based on the 24h grid fluctuations of a typical day in winter and summer in India in 2018/19 (refer to Figure S4 for the hourly grid fluctuations in India and the impacts on the carbon intensity of electricity). Daytime charging starts at 8 am, while overnight charging starts at 8 pm. Charging duration estimated based on the BEV's battery size and the use of a typical 2.5 kW charger [32], with a total vehicle lifetime distance of 200,000km.

Line 112: “Correspondingly, it allows BEVs in 2030 to reduce their life cycle emissions by 23-25% (Figure S13) relative to a BEV in the year 2018/19”. This is true under these assumptions for a BEV purchased in 2019 and also driven in 2030. But a future BEV purchased in 2030 will have a cleaner grid and also be more efficient to drive (lower Wh/km), due to technology improvement, competition, and potentially regulation. Need to make this clear in the text and calculations.

This is a major point that has been presented in the subsequent sentences and paragraphs (Line 112-137). For example, in Figure 3, we demonstrated the GHG reduction potentials of the BEV as a result of improving the efficiency of the BEV. This is then followed by discussions on the opportunity that comes with a more rapid reduction in the grid carbon intensity as shown in Figure 4.

However, to add more clarity, we included the following text as per reviewer suggestion:

“Correspondingly, it allows BEVs in 2030 to reduce their life cycle emissions by 23-25% (Figure S15) relative to a BEV in 2018/19. A more rapid reduction in the electric grid carbon intensity could enable a larger reduction in the life cycle GHG emissions of the BEV. Additionally, the climate change mitigation potential for a BEV in 2030 also depends on its electricity consumption for every kilometer driven (i.e., Wh/km) and the fuel efficiency of the combustion engine vehicle baseline (i.e., L/100km) for that year (Figure 3).”

Line 115: “Phase 2 of the fuel consumption standard in India, which comes into effect from the year 2022/23 onwards [30], stipulates that 4W passenger vehicles with an average curb weight of 1237 kg should not exceed 4.95L/100km. Its overall GHG emissions would be comparable to a BEV with an energy consumption of 142 Wh/km (Figure 3) for a gasoline spark-ignition vehicle.” Two of the three existing 4Ws in India listed by the authors in Table 1 have energy consumption values of well below 143 Wh/km (100 and 101). Energy efficiency of BEVs is also potentially improved in the future as per my previous comment. In addition “it’s overall GHG emissions” would not be comparable at this value- it’s GHG emissions for that year for that specific grid carbon intensity, which is likely to continue to fall over the many years of the vehicle’s life.

There are very limited BEVs in India today. Our selection of BEVs are based on what is available in the market today. For sure, there will be more BEV models entering the Indian market by 2030. The electricity consumption per km driven (Wh/km) may not be as low as the models available today since the 3 BEV models used in this study are small and basic vehicles. Experiences in other countries (e.g. China) show that larger BEV models will become available over time. Therefore, the average electricity consumed per vehicle km depends not just on the efficiency improvement but also on the structural shift in consumer preferences. India, like many other countries globally, is experiencing a gradual shift towards larger passenger cars, with SUVs accounting for a significant 34% of the total vehicle sales in India in 2019. If the structural shift happens faster than the improvements in vehicle efficiency, then the average electricity consumption per km will increase. Furthermore, it is important to note that an electric powertrain already has a very high efficiency of more than 80%, therefore it is reasonable to expect a diminishing marginal gain in efficiency over time.

Predicting the average Wh/km of the BEV in 2030 is a complex task that will involve a consumer choice modelling – this is beyond the scope of our work, and there isn’t enough data for such a detailed assessment. Therefore, we think it is more appropriate to conduct a sensitivity analysis, as depicted in

Figure 4, with varying BEV electricity consumption and contrasted against gasoline vehicles with 3 different fuel consumption values.

Figure 3: Does this model how the grid changes over the course of the vehicle’s life? If not, the Figure does not support the conclusions. What are the assumptions about what time of day the vehicle is charged? Are these life cycle emissions? Need clarification in the caption.

There are 3 main parameters for this plot: grid intensity, BEV electricity consumption (Wh/km) and the fuel consumption (L/100km) of the gasoline vehicle comparator. In Figure 3, we fixed the grid intensity, and varied the other 2 parameters. However, this is complemented by 2 sensitivity analyses: (a) in Figure S15b, we also varied the grid carbon intensity; and (b) more importantly, in Figure 4 we ran a more aggressive grid decarbonization scenario. In Figure 4, we plotted the grid carbon intensity as an independent variable (x-axis) so that the readers can estimate the GHG reduction potentials of the BEVs based on any projected grid mix (and speed of decarbonization over time).

As per suggestion from the reviewer, we have clarified in the caption of Figure 3 that we refer to life cycle emissions, and the generation mix is based on the annual, national average in 2030. See below:

Figure 3 Contour plot mapping the life cycle GHG emission reduction (in percentage) of BEVs with a cleaner power future. The availability of cleaner electricity in the year 2030 can enable larger GHG reduction for BEVs. The percent GHG reduction depends on the efficiency of future BEVs and the baseline gasoline vehicle in 2030. The energy generation mix in 2030 is based on the national annual average. It is assumed to be static, with additional sensitivity plots in Figure S15 (b) of the supporting document and in Figure 4 below.

Figure 3: at editor’s discretion: change kerb to curb for consistency throughout – We changed to curb

weight for overall consistency.

Figure 4: This figure shows GHG reduction potentials for all BEVs for any grid cleaner than 500 g/kWh. So doesn't it show that BEVs have a mitigation potential and contradict the message of the paper?

We agree with the reviewer that BEVs can have large GHG reduction potentials when powered by cleaner grid electricity. We have stated this throughout the manuscript, for example in line 128-129: *“Limiting the global temperature increase to below 1.5°C will require almost complete decarbonization of the power sector by the mid-century. Progress towards this target will lower emissions of BEV, resulting in substantial GHG benefits for the passenger road transport sector (Figure 4).”*

We moderated the language throughout the manuscript accordingly, and in particular in the conclusions section to make sure that it is consistent with the findings of this study.

Line 138: “gasoline-electric and diesel-electric hybrids” it looks like these are plug-in hybrid electric vehicles (PHEVs) rather than hybrid-electric vehicles (HEVs) which do not plug in and are mentioned later in the sentence, and need to be denoted as PHEVs, and the all-electric range stated. However in the next sentence, it says that no additional infrastructure is required for PHEVs, which is not the case.

This refers to HEV, and not plug-in hybrids (PHEV). There were no PHEVs available in the Indian market in 2018/2019, and therefore we did not include it in our analysis.

Line 144: This conclusion is only for an electric grid that does not get cleaner

In line 44, we recommended that policies also encourage the development and uptake of HEVs. This recommendation is not exclusive and/or at the expense of our wider recommendation for accelerating grid decarbonization to ensure significant GHG reduction potentials for BEVs.

In the best-case scenario where the grid rapidly decarbonizes, and BEVs account for 30% of new vehicle sales (as per the EV30@30 program), there is still a large majority of vehicles that will continue to be powered by combustion engines. It is better that these vehicles are hybridized since it will contribute towards mitigating emissions from the road transport sector through efficiency improvements. This is in-line with the avoid-shift-improve framework. In the worst-case scenario, where the grid carbon intensity doesn't improve, or even worsens, then HEV stands a better chance to reduce transport emissions. Therefore, we think that the recommendation is defensible especially when read in the context of the entire paragraph: *“In contrast, BEVs’ real GHG mitigation potential will critically rely on the speed of grid decarbonization through investments in cleaner power and the widespread deployment of charging networks. Given the power mix in India in 2019, BEVs offer smaller GHG reduction potential and with higher uncertainty due to geographical and temporal variabilities (Figure S8). It is, therefore, prudent for policies to **also** encourage rapid development and uptake of advanced hybrid solutions with varying electrification levels.”*

Line 186: please clarify how a 2W removes range anxiety or remove that part of the sentence. We removed this as per reviewer suggestion.

Line 191: vehicle sales - Revised as per reviewer suggestion.

Line 190: The entire critical materials section is very well done. Nice job. Thank you.

Line 232: define NEDC. Revised as per reviewer suggestion..

Line 249: I don't know if it's a mistake, rather than an undercounting- there could be innovation, manufacturing, or air pollutant avoidance reasons for favoring EVs in tailpipe regulations. There are several papers about how existing fuel economy regulations in various countries undercount EVs in their fuel economy regulations. Those should be cited here.

We have removed the word "mistaken" and added several references as suggested by the reviewer:

"Most fuel economy and vehicle emission standards worldwide exempt the GHG consequences of upstream emissions, including power generation. While several preferential treatments were originally designed into regulations to jump-start the BEV industry [62, 63, 64], its continued and widespread use risk undermining efforts to mitigate GHG emissions from road transport globally [61]."

Line 268: "4W BEVs do not offer many prospects for reducing transport GHG in India within the timeframe of this study" this is not supported by the authors' calculations and results.

This is a fair comment and we have revised the text accordingly, as per the following:

"However, with the existing electricity mix, 4W BEVs do not offer a significant reduction in transport GHGs in India within the timeframe of this study. To improve the prospect for BEVs, India must integrate its mobility transition plan into a broader framework to ensure that its implementation leads to a deep reduction in overall GHG emissions."

Line 294: finish sentence. We have corrected this sentence as per reviewer suggestion.

Line 295: "highly efficient combustion engine-powered vehicles, and low-carbon fuels such as biofuels, hydrogen, and CO₂-derived fuels, including Methanol and DME". All of these fuels currently, even when CCS is used (which is not yet in use), generate some or a substantial amount of CO₂ and are not a zero carbon solution. The reviewer is correct that all the above solutions are not zero emissions today. But, likewise, our studies (and many others) have shown that BEVs are also not zero emissions given the power generation mix today. The prospect for deep emission reduction in a BEV depends primarily on how the electricity is generated. The same requirement applies to, for example electro-fuels, where the electricity has to be based on low-carbon renewable electricity. Similarly, the emissions intensity of hydrogen will depend on how it is produced – electrolytic hydrogen production also requires low-carbon electricity. On the hand, for biofuels, land-use and agricultural practices matters a lot to ensure that they are sustainable. This goes to show that as we transition to alternative fuels and vehicles, the scope of emissions regulations needs to evolve from a simple tailpipe measurement to a life cycle-based emissions regulation to ensure that our transition is on a trajectory towards net zero.

Methods: The authors' values used for a Tata Nexon and 927 g CO₂/kWh in the current Indian grid generate direct CO₂ values of 93 g/km for the EV, 114 g/km for the gasoline vehicle, and 118 g/km for the diesel vehicle. Even with this very high electricity carbon intensity that will come down, the EV is the

best choice (from a direct perspective, and upstream values will add some but unlikely flip this answer as both EVs and petrol have upstream impacts). The paper is written and the conclusions stated that 4Ws are worse or negligible.

In the example above, the reviewer has not taken into consideration of the real-world driving correction factor. Real-world driving can reduce the total electric range of a BEV by 30%, as reported by many researchers. A 30% reduction in electric range translates to more than 40% energy consumption penalty per km of BEV driven. On top of that, the reviewer hasn't included the emissions associated with vehicle manufacturing, which is relatively larger for BEVs given the emissions associated with battery manufacturing.

Line 370: the authors used a 40% degradation of real world fuel economy for BEVs, referring to Argonne National Laboratory and citing several older reports and conference papers. This 40% number is inconsistent with most BEV studies. Are these citations for PHEVs? This assumption alone has the ability to change the answer and sensitivity analysis is needed.

The references we used are relatively new, which include from reputable peer-reviewed journals. There are many others in the literature that report real-world penalties of about 40%. The more recent literature that the reviewer might be referring to may have reported a smaller deviation, however, it is important to make sure that the comparison is done relative to the NEDC drive cycle. The more recent literatures tend to compare real-world driving against the newly adopted WLTC test procedure (or even against the US EPA test cycles). The WLTC test procedure, which is now widely used in the EU, or even the more stringent US EPA test cycles, are designed to minimize the deviation between vehicle certification and real-world performances, hence the penalty factors are smaller. In India, the official type approval is still based on the outdated NEDC test cycle and therefore we have decided to use the older correction factors. However, for completeness, we have conducted additional sensitivity analysis (Figure 9) as per the reviewer's recommendation. We have also added a new section in the report, called "Reflecting on the limits of generalizations" to discuss the implications of our methodological choices and assumptions. Please find below the discussion pertaining to the real-world correction factor:

"Second, India's fuel/energy consumption certification is based on the New European Driving Cycle (NEDC) and, to the best of our knowledge, there isn't an industry-standard correlation between the type-approved vehicle certifications and real-world driving characteristics specific to India. The NEDC is known to significantly underestimate the real-world emissions and energy consumptions of vehicles, with some reporting larger penalties for heavily electrified vehicles [75]. Thus, in this study, we adopted correction factors consistent with the methods in recent literature [10, 76, 77, 78, 79, 6, 75] to adjust the certified energy/fuel consumptions to reflect real-world deteriorations. There is large uncertainty and implication with this assumption (Figure S11); however, removing the correction factors (as done in Figure S12) ignores the real-world effects that have been well-documented in the literature [80, 81, 82, 83, 84, 85]. In a 2021 report, the International Council on Clean Transportation (ICCT) adopted a real-world penalty of 34% for conventional combustion engine vehicles, while the real-world electric range of BEVs would deteriorate by as much as 30% based on empirical evidence [86]. Figure 9 depicts the total life cycle GHG emissions of BEVs, gasoline, and diesel vehicles using the ICCT's real-world correction factors. While this resulted in slight changes to individual powertrains' relative GHG reduction merits, it doesn't alter the overall findings."

Figure 9 The effects of adopting an alternative set of real-world correction factors on the life cycle GHG emissions of electric, gasoline, and diesel vehicles. Compared to the NEDC, the electric range of a BEV is reduced by 30%, while the fuel consumption of combustion engine vehicles increased by 34% [86]. Total vehicle lifetime distance of 200,000km was assumed.

Line 385: the assumption of 124.5 kg/kWh of BEV battery manufactured is at the upper end of the distribution of values in the literature, and about double the values used by Argonne National Laboratory. What happens if lower values are used/achieved?

The value adopted in our study was obtained from a recent scientific publication in 2020, based on actual, primary data in China (<https://doi.org/10.1016/j.jclepro.2020.123006>). As the reviewer might be aware, China accounts for the largest battery manufacturing share in the world. The value quoted by the reviewer (124.5 kgCO₂eq/kWh) is the battery manufacturing carbon intensity. When the end-of-life recycling credits are included, the net emissions intensity drops to 93.6kgCO₂eq/kWh. This is not too different from a 2019 report by Argonne (see below). Furthermore, battery manufacturing accounts for about 5% of the total life cycle GHG emissions (Figure S19 & S20). Thus, the life cycle emission is unlikely to change substantially with slight alterations in battery manufacturing emissions.

Source: Kelly, Dai & Wang, Argonne National Laboratory, 2019

Line 390: please provide a reference for the recycling credit assumption. We have added the reference as per reviewer suggestion, thank you.

Line 410: "The excel models, with data sources, are available from the corresponding author upon reasonable request." For a paper like this, in order to be published all of these models need to be publicly available online with a DOI.

We are willing to make the models available, and have communicated this to the Editor.

Reviewer #2 (Remarks to the Author):

I have gone through the revisions made by the authors and satisfied with the revisions made to the paper

We wish to thank the reviewer for all the comments and feedback.

Reviewer #3 (Remarks to the Author):

The revised manuscript provides clarity and the reviewer has enjoyed reading the article. The data provided and methodology is clearly stated such that the work can be replicated by others. The authors have provided proper justifications to the queries raised in the previous review. However, the reviewer requires justification for following

We wish to thank the reviewer for the time and effort taken to provide constructive feedbacks on our manuscript.

1. The Figure.2 describes the hourly fluctuations of GHG emission for charging a vehicle in winter and summer. Please provide the details of charging considered, i.e., whether the battery is charged by a level-1, level-2 or level-3 charger. For example, if a 10kWh battery is charged by a level-1 charger of 1kW, the energy consumed from the grid is around 1kWh over the period of 10 hour. This would reduce the GHG emission during that window of the charging when compared to level-3 charging that utilises total 10kWh of energy in an hour, thereby increasing the emissions.

We completely agree with the reviewer. We have recalculated the life cycle emissions and replaced Figure 2 with a new plot, which now takes into account of the BEV time-of-charge and the charging duration. The latter is estimated from the battery size and the power of the charger. We thank the reviewer for this good observation and feedback. See below for the updated plot.

Figure 2 Effects of battery recharging time on the life cycle GHG emissions of BEVs based on the 24h grid fluctuations of a typical day in winter and summer in India in 2018/19 (refer to Figure S4 for the hourly grid fluctuations in India and the impacts on the carbon intensity of electricity). Daytime charging starts at 8 am, while overnight charging starts at 8 pm. Charging duration estimated based on the BEV's battery size and the use of a typical 2.5 kW charger [32], with a total vehicle lifetime distance of 200,000km.

2. The author is suggested to also consider the emissions during manufacturing of EV and the emissions during manufacturing of replacements components in EV required due to faults over its lifetime.

This study has included the emissions associated with vehicle and components manufacturing, as per data extracted from Sphera's LCA software and databases (GaBi version 9.2 with 2020 LCI Databases). We thank the reviewer for the suggestion.

3. The authors have mentioned 200,000 kms as life time for Four wheeler but no proper referencing of the source is provided. Justify how authors came to conclusion of 200,000 kms as life of EV. We thank the reviewer for pointing this out, and we have added further clarification with appropriate references in the text.

"As India does not yet have a robust vehicle scrappage policy, this study assumed a total vehicle lifetime travel distance of 200,000 km (i.e., 12,500 annual kilometers [98, 99] over 16 years), roughly in-line with other studies [86]."

Moreover, as we varied the cumulative lifetime mileage by +/- 10% (i.e., 180,000 – 220,000km), the total life cycle GHG emissions changed by less than +/- 1%, as shown in Figure S11 in the supporting document.

4. The author mentions "The life cycle emissions of BEVs oscillate over a wide range depending on when the charging takes place." Provide justification.

The GHG emissions of BEVs fluctuate depending on when it is charged given the temporal variability in the power generation mix as shown in Figure S4 of the Supporting document. This is largely linked to the intermittencies of renewables, which tends to peak during mid-day. We have added references to Figure 2 and Figure S4 in the revised manuscript to make this link more explicit.

"The life cycle emissions of BEVs oscillate over a wide range depending on when the charging takes place (Figure 2 and Figure S4). Off-peak, overnight charging in India results in larger GHG emissions than daytime charging – a difference of 3-9%."

Reviewer comments, second review -

Reviewer #1 (Remarks to the Author):

Overall: The authors have put a lot of great work into this article and the revised version is considerably improved. There are a few remaining stylistic things needed before publication. This article conducts a regional LCA of EVs light-duty vehicles and motorscooters in India and makes an important contribution. The paper as currently written says that EVs in India don't reduce GHGs that much, because of the amount of coal in the grid in India, and states that EVs are not the best solution for India, and alludes instead to a mosaic of options. One of those options nicely evaluated in the paper are EV motorscooters. Demand reduction is necessary and is mentioned but not modeled, which is OK. But all the other technical options listed (CO₂-derived fuels, fuel cells largely rely on the same zero-carbon electricity grid needed, and are not currently commercially viable, do not have infrastructure or in some cases are still in the lab phase. The conclusion of the article could have more impact if it identified the pace and scale of electricity carbon intensity reduction in India needed, in order for the EV30@30 plan to have a large GHG reduction. India has a plan to reduce the carbon intensity of electricity as well as reduce CO₂ from transport. The results of this LCA can help guide stakeholders on the performance targets needed for both of these sectors to be successful.

Title: suggest considering a title such as

Electrifying passenger road transport in India requires near-term electricity grid decarbonization

Life cycle GHGs of passenger road transport in India favors electric 2-wheelers as a robust solution

Line 5: "Here we demonstrate BEVs as the complement and not the panacea for passenger road transport in India." This is the research question evaluated- what are the complements that were evaluated in the paper besides EV motorscooters that would eventually bring transport GHGs substantially down? Diesels and HEVs do not provide enough GHG reductions. So need to be explicit what the complements are.

Line 26: consider changing in-use to use phase

Line 72: "Climate change responses for the transport sector necessitate a balanced mix of behavioral modifications and technology-oriented policies with diversity across the three dimensions - time, region, and sectors. Although climate change is a global challenge, there is no universal, one-size-fits-all technology solution and policy solution." It would be good to add one more sentence here to elaborate. While demand reduction through public transit, urban design, and policy is essential, the sentence, and the overall message of the paper seems to allude to other technologies. If the argument is that non-Annex-1 countries should not bring their transportation and electricity sectors to close to zero emissions by mid-century, then clearly state that. But if the argument that India can decarbonize the transportation sector with a technology other than electric vehicles, then make that argument. Right now, the paper does a good job showing that under current and near-term electricity mixes, in some regions EVs do not result in considerable GHG reductions. On Line 140, the paper argues for HEVs, which do reduce GHGs, but since vehicles last on the road for a long time, locks in this technology and doesn't encourage the building of charging infrastructure. A HEV-focused strategy would make a contribution, but in 2040, it would be very difficult to then start up an EV focused strategy and get to net-zero by 2050 or 2060. HEVs would also leave a lot of the on-road air pollutants in place. The argument made for EV 2W on Line 182 is very good and potentially should be woven into the abstract and this intro.

Line 86: "The climate change mitigation potential of a BEV is particularly diminished when compared to a diesel engine, given the latter's higher efficiency." A diesel engine is more efficient

than a gasoline engine but is not more efficient than an electric motor, so a reader might take the wrong message from this. For a given EV Wh/km consumption and regional electricity grid CO₂/kWh, there is a diesel vehicle fuel efficiency that will result in the same or lower GHG/km. What is this fuel efficiency number and do diesel vehicles exceed it India? It's in the tables and SI and in Figure 8 but it's worth mentioning the range of l/100km here in the main text here. Alternatively reword the sentence to say that depending on the grid GHG intensity, high efficiency petrol vehicles such as diesels or HEVs can approach or be lower than the CO₂/km values of EVs.

Line 264: "However, with the existing electricity mix, 4W BEVs do not offer a significant reduction in transport GHGs in India within the timeframe of this study." This is likely going to be the conclusion used by others for this study, so it requires some context. Instead of saying 'significant reduction', please be quantitative in this sentence- what is the percentage ranges of reductions? Can the years of this study be re-stated in this study? Also, can you provide a goal here to help guide decisionmakers? Such as "Electricity CO₂ must be less than XXX g/kWh in order for EVs to make a XX% reduction in GHGs relative to gasoline vehicles".

Line 267: this is very important about 2W- can this go in the abstract?

Line 278: again, what are the elements of a bespoke policy mix here? It sounds like there is a good case to be made for electrified 2Ws- can that be said here? Or is there something else that you wanted to suggest besides electric 4W?

Line 291: "Thus, we need a mosaic of technologies for different transport sectors and geographical regions, including electrified powertrains (counting full BEV), highly efficient combustion engine-powered vehicles, and low-carbon fuels such as biofuels, hydrogen, and CO₂-derived fuels, including Methanol and DME." This statement is not in line with net zero emissions from the transport sector by 2050. HEVs and diesels still generate a lot of GHGs. Low-carbon/low-impact biofuels do not yet exist at scale. H₂ vehicles need H₂ made from clean electricity to be ultra-low carbon, and if a country has a lot of clean electricity, it is more efficient just to power EVs than to convert to H₂ and power a FCEV (which do not exist commercially at scale). CO₂-derived fuels need a clean electricity system and do not exist commercially. In order to make the argument that any of these other solutions are needed, a statement recognizing that any light-duty transport sector not using EVs will therefore delay decarbonization to some date in the future, using technologies that are not yet commercially viable and widely available. If that is the argument, make that clear. Otherwise, a reader might assume these other technologies are available now and are not reliant on the same clean electricity system needs of EVs.

Reviewer #2 (Remarks to the Author):

I am satisfied with the revisions to the paper.

However, I would suggest that the authors revise the abstract to remain closer to the findings and avoid broad generalisations e.g., In ln6 , it is not clear as to what BEVs complement. In ln 10 60% increase in central regions however, in the paper, there is no central region.

Reviewer #3 (Remarks to the Author):

I Have gone through the manuscript and it has been substantially improved and is acceptable for publication

RESPONSES TO REVIEWER’S COMMENTS

Reviewer #1:

Overall Comment 1: The authors have put a lot of great work into this article and the revised version is considerably improved. There are a few remaining stylistic things needed before publication. This article conducts a regional LCA of EVs light-duty vehicles and motorscooters in India and makes an important contribution. The paper as currently written says that EVs in India don’t reduce GHGs that much, because of the amount of coal in the grid in India, and states that EVs are not the best solution for India, and alludes instead to a mosaic of options. One of those options nicely evaluated in the paper are EV motorscooters. Demand reduction is necessary and is mentioned but not modeled, which is OK. But all the other technical options listed (CO₂-derived fuels, fuel cells largely rely on the same zero-carbon electricity grid needed, and are not currently commercially viable, do not have infrastructure or in some cases are still in the lab phase. The conclusion of the article could have more impact if it identified the pace and scale of electricity carbon intensity reduction in India needed, in order for the EV30@30 plan to have a large GHG reduction. India has a plan to reduce the carbon intensity of electricity as well as reduce CO₂ from transport. The results of this LCA can help guide stakeholders on the performance targets needed for both of these sectors to be successful.

Response to the Reviewer: We thank the reviewer for taking the time to review our manuscript and provide constructive feedback and suggestions to improve the quality and readability of our manuscript. We have considered all of the reviewer’s suggestions below in the updated manuscript. In particular, we have revised the abstract to make it more concise and insightful, and we have also provided our views on how India’s transport and power sector should evolve in the near future. Below, we provide point-by-point responses to each of the reviewer’s suggestions.

Comment 2: Title: suggest considering a title such as
Electrifying passenger road transport in India requires near-term electricity grid decarbonization
Life cycle GHGs of passenger road transport in India favors electric 2-wheelers as a robust solution

Response to the Reviewer: The reviewer has suggested two good titles. We have updated the manuscript title to the following: “Electrifying passenger road transport in India requires near-term electricity grid decarbonisation”

Comment 3: Line 5: “Here we demonstrate BEVs as the complement and not the panacea for passenger road transport in India.” This is the research question

evaluated- what are the complements that were evaluated in the paper besides EV motorscooters that would eventually bring transport GHGs substantially down? Diesels and HEVs do not provide enough GHG reductions. So need to be explicit what the complements are.

Response to the Reviewer: We agree with the reviewer that this sentence raises more questions. The journal word count limits us for the abstract (150 words max), and therefore we have removed this sentence completely and re-written the abstract. See below for the revised abstract (current word count 150 words):

“Battery-electric vehicles (BEV) have emerged as a favoured technology solution to mitigate transport greenhouse gas (GHG) emissions in many non-Annex 1 countries, including India. GHG mitigation potentials of electric 4-wheelers in India depend critically on when and where they are charged: 40% reduction in the north-eastern states and more than 15% increase in the eastern/western regions today, with higher overall GHGs emitted when charged overnight and in the summer. Self-charging gasoline-electric hybrids can lead to 33% GHG reductions, though they haven’t been fully considered a mitigation option in India. Electric 2-wheelers can already enable a 20% reduction in GHG emissions given their small battery size and superior efficiency. India’s electrification plan demands up to 125GWh of annual battery capacities by 2030, nearly 10% of projected worldwide productions. India requires a phased electrification with a near-term focus on 2-wheelers and a clear trajectory to phase-out coal-power for an organised mobility transition.”

Comment 4: Line 26: consider changing in-use to use phase.

Response to the Reviewer: Corrected as per reviewer’s suggestion.

Comment 5: Line 72: “Climate change responses for the transport sector necessitate a balanced mix of behavioral modifications and technology-oriented policies with diversity across the three dimensions - time, region, and sectors. Although climate change is a global challenge, there is no universal, one-size-fits-all technology solution and policy solution.” It would be good to add one more sentence here to elaborate. While demand reduction through public transit, urban design, and policy is essential, the sentence, and the overall message of the paper seems to allude to other technologies. If the argument is that non-Annex-1 countries should not bring their transportation and electricity sectors to close to zero emissions by mid-century, then clearly state that. But if the argument that India can decarbonize the transportation sector with a technology other than electric vehicles, then make that argument. Right now, the paper does a good job showing that under current and near-term electricity mixes, in some regions EVs do not result in considerable GHG reductions. On Line 140, the paper argues for HEVs, which do reduce GHGs, but since vehicles last on the road for a long time, locks in this technology and doesn’t encourage the building of charging infrastructure. A HEV-focused strategy would make a contribution, but in 2040, it would be very difficult to then start up

an EV focused strategy and get to net-zero by 2050 or 2060. HEVs would also leave a lot of the on-road air pollutants in place. The argument made for EV 2W on Line 182 is very good and potentially should be woven into the abstract and this intro.

Response to the Reviewer: We have expanded this section slightly to elaborate this point further, as per the reviewer's suggestion. Please find below the updated section:

“Climate change responses for the transport sector necessitate a balanced mix of behavioural modifications and technology-oriented policies with diversity across the three dimensions - time, region, and sectors. Although climate change is a global challenge, there is no universal, one-size-fits-all technology solution and policy solution. Ultimately, India's EV30@30 ambition must be complemented by a near-term commitment to phase-out coal from the power sector to improve the GHG reduction prospect of a 4W BEV. In the absence of a clear trajectory to phase down coal, the electrification of transport should prioritise India's growing 2-wheel segment that could still offer about a 20% GHG benefit. This has to be complemented by a more rigorous fuel efficiency standard and low-carbon fuels standard to drive the adoption of highly-efficient 4W engines (e.g., hybridised vehicles) and fuels with a lower climate impact (e.g., sustainable low-carbon fuels) as a transitional solution.”

Comment 6: Line 86: “The climate change mitigation potential of a BEV is particularly diminished when compared to a diesel engine, given the latter's higher efficiency.” A diesel engine is more efficient than a gasoline engine but is not more efficient than an electric motor, so a reader might take the wrong message from this. For a given EV Wh/km consumption and regional electricity grid CO₂/kWh, there is a diesel vehicle fuel efficiency that will result in the same or lower GHG/km. What is this fuel efficiency number and do diesel vehicles exceed it India? It's in the tables and SI and in Figure 8 but it's worth mentioning the range of l/100km here in the main text here. Alternatively reword the sentence to say that depending on the grid GHG intensity, high efficiency petrol vehicles such as diesels or HEVs can approach or be lower than the CO₂/km values of EVs.

Response to the Reviewer: We thank the reviewer for picking this up. We agree that the way it was written can be confusing and may imply that diesel has higher efficiency than a BEV. The reviewer is correct in pointing that an electric powertrain is much more efficient than a conventional engine. We have reworded this sentence as follows:

“The climate change mitigation potential of a BEV is particularly diminished compared to a diesel vehicle, given that the latter has ~25% lower fuel consumption than an equivalent gasoline vehicle (Table 1).”

Comment 7: Line 264: “However, with the existing electricity mix, 4W BEVs do not offer a significant reduction in transport GHGs in India within the timeframe of this study.” This is likely going to be the conclusion used by others for this study, so it requires some context. Instead of saying ‘significant reduction’, please be quantitative in this sentence- what is the percentage ranges of reductions? Can the years of this study be re-stated in this study? Also, can you provide a goal here to help guide decision makers? Such as “Electricity CO2 must be less than XXX g/kWh in order for EVs to make a XX% reduction in GHGs relative to gasoline vehicles”.

Response to the Reviewer: We have updated the manuscript as per the reviewer’s suggestion to provide a quantitative GHG reduction overview and assess the electricity carbon intensity reductions required in the near term. See below for the updated section:

“There is growing policy interest in BEVs in India – a highly populated subcontinent with an intense appetite for personal mobility. However, with the existing electricity mix, at best, 4W BEVs offer less than 5% GHG reduction compared to the sales-weighted average vehicle sold in India in 2018/2019 (comprising 65% gasoline and 35% diesel) (Figure S22). By 2030, the GHG reduction potential of a 4W BEV depends critically on how quickly coal can be phased-out from the power generation mix. To offer at least a 20% reduction in overall GHGs, the electricity carbon intensity has to be less than 500 gCO₂eq/kWh, or well below 350 gCO₂eq/kWh to attain more than 50% GHG reduction (Figure S22). This is 46% and 62% below the current electricity carbon intensity level, implying a near-term and rapid phasing-out of coal from the power sector, which is an unlikely timeline for the EV30@30 ambition. Given India’s complicated relationship with the coal industry, even the phasing-down of its use is a large undertaking, as seen in the recently concluded 26th Conference of Parties (COP26) at Glasgow [66].”

		(a) BEV - 1225kg, 21.2 kWh (equivalent to Mahindra eVerito)																	
		Carbon intensity of electricity - gCO _{2eq} /kWh																	
		100	150	200	250	300	350	400	450	500	550	600	650	700	750	800	850	900	
Electricity consumption (Wh/km)	100	42	51	59	68	77	86	94	103	112	121	129	138	147	155	164	173	182	
	110	44	53	63	73	82	92	101	111	121	130	140	149	159	168	178	188	197	
	120	45	56	66	77	87	98	108	119	129	140	150	161	171	182	192	203	213	
	130	47	59	70	81	93	104	115	127	138	149	161	172	183	195	206	217	229	
	140	49	61	73	86	98	110	122	134	147	159	171	183	196	208	220	232	244	
	150	51	64	77	90	103	116	129	142	155	168	182	195	208	221	234	247	260	
	160	52	66	80	94	108	122	136	150	164	178	192	206	220	234	248	262	276	
	170	54	69	84	99	114	128	143	158	173	188	203	217	232	247	262	277	292	
	180	56	72	87	103	119	134	150	166	182	197	213	229	244	260	276	292	307	
	190	58	74	91	107	124	141	157	174	190	207	223	240	257	273	290	306	323	
	200	59	77	94	112	129	147	164	182	199	216	234	251	269	286	304	321	339	

		(b) BEV - 1400kg, 30.2 kWh (equivalent to Tata Nexon EV)																	
		Carbon intensity of electricity - gCO _{2eq} /kWh																	
		100	150	200	250	300	350	400	450	500	550	600	650	700	750	800	850	900	
Electricity consumption (Wh/km)	100	47	56	65	73	82	91	99	108	117	126	134	143	152	161	169	178	187	
	110	49	58	68	78	87	97	106	116	126	135	145	154	164	174	183	193	202	
	120	51	61	72	82	92	103	113	124	134	145	155	166	176	187	197	208	218	
	130	52	64	75	86	98	109	120	132	143	154	166	177	188	200	211	222	234	
	140	54	66	79	91	103	115	127	140	152	164	176	188	201	213	225	237	250	
	150	56	69	82	95	108	121	134	147	161	174	187	200	213	226	239	252	265	
	160	58	72	86	99	113	127	141	155	169	183	197	211	225	239	253	267	281	
	170	59	74	89	104	119	133	148	163	178	193	208	222	237	252	267	282	297	
	180	61	77	92	108	124	140	155	171	187	202	218	234	250	265	281	297	312	
	190	63	79	96	113	129	146	162	179	195	212	229	245	262	278	295	311	328	
	200	65	82	99	117	134	152	169	187	204	222	239	257	274	291	309	326	344	

Figure S22 Life cycle GHG emissions of 4W BEVs with different electricity consumptions and powered by varying electricity carbon intensity for (a) a BEV with 1225kg curb weight and equipped with a 21.2 kWh battery, equivalent to a Mahindra eVerito, and (b) a BEV with 1400kg curb weight and equipped with a 30.2 kWh battery, equivalent to a Tata Nexon EV. The sales-weighted average vehicle in India in 2018/19, comprising of 65% gasoline and 35% diesel cars, has a curb weight of 1078 kg and tailpipe emission of 121.9 gCO₂/km [12], resulting in an estimated total life cycle GHG emission of 197gCO_{2eq}/km. The current grid mix has a carbon intensity of 927 gCO_{2eq}/kWh.

Comment 8: Line 267: this is very important about 2W- can this go in the abstract?

Response to the Reviewer: We agree with the reviewer. This is now woven into the abstract, as per the following:

“Battery-electric vehicles (BEV) have emerged as a favoured technology solution to mitigate transport greenhouse gas (GHG) emissions in many non-Annex 1 countries, including India. GHG mitigation potentials of electric 4-wheelers in India depend critically on when and where they are charged: 40% reduction in the north-eastern states and more than 15% increase in the eastern/western regions today, with higher overall GHGs emitted when charged overnight and in the summer. Self-charging gasoline-electric hybrids can lead to 33% GHG reductions, though they haven’t been fully considered a mitigation option in India. Electric 2-wheelers can already enable a 20% reduction in GHG emissions given their small battery size and superior efficiency. India’s electrification plan demands up to 125GWh of

annual battery capacities by 2030, nearly 10% of projected worldwide productions. India requires a phased electrification with a near-term focus on 2-wheelers and a clear trajectory to phase-out coal-power for an organised mobility transition.”

Comment 9: Line 278: again, what are the elements of a bespoke policy mix here? It sounds like there is a good case to be made for electrified 2Ws- can that be said here? Or is there something else that you wanted to suggest besides electric 4W?

Response to the Reviewer: The reviewer is right that we are referring to the electrification of 2W and the need to focus on near-term phasing-out of coal to improve the GHG reduction prospects of 4W BEVs (this is already elaborated in the preceding paragraphs). We also refer to the need for a wider transport decarbonisation plan that considers the “avoid-shift-improve” framework. These are discussed in the subsequent paragraph, as per below:

“Technology-centric policy on its own is not adequate to meet the sectoral mitigation targets [5], and the ‘avoid-shift-improve’ framework offers the best chance for achieving sustainable mobility [69, 70]. Effective behaviour-centric policies are needed to attenuate demand for road vehicle travel [4] through (i) support for active mobility (cycling and walking), (ii) enhancement of public transit, (iii) optimal design of the built environment, and (iv) reducing the need for travel (telecommute and telework).”

Comment 10: Line 291: “Thus, we need a mosaic of technologies for different transport sectors and geographical regions, including electrified powertrains (counting full BEV), highly efficient combustion engine-powered vehicles, and low-carbon fuels such as biofuels, hydrogen, and CO₂-derived fuels, including Methanol and DME.” This statement is not in line with net zero emissions from the transport sector by 2050. HEVs and diesels still generate a lot of GHGs. Low-carbon/low-impact biofuels do not yet exist at scale. H₂ vehicles need H₂ made from clean electricity to be ultra-low carbon, and if a country has a lot of clean electricity, it is more efficient just to power EVs than to convert to H₂ and power a FCEV (which do not exist commercially at scale). CO₂-derived fuels need a clean electricity system and do not exist commercially. In order to make the argument that any of these other solutions are needed, a statement recognizing that any light-duty transport sector not using EVs will therefore delay decarbonization to some date in the future, using technologies that are not yet commercially viable and widely available. If that is the argument, make that clear. Otherwise, a reader might assume these other technologies are available now and are not reliant on the same clean electricity system needs of EVs.

Response to the Reviewer: We agree with the reviewer that these technologies are only low-carbon if produced under the right context and environment. We elaborated this section to provide further clarification. See below:

“Thus, we need a mosaic of technologies for different transport sectors and geographical regions, including electrified powertrains (counting full BEV), highly efficient combustion engine-powered vehicles, and low-carbon fuels such as biofuels, hydrogen, and CO₂-derived fuels, including Methanol and DME. However, these too can only thrive in a sustainable and low-carbon ecosystem, including the accessibility to affordable and clean power, the adoption of improved land-use and sustainable agricultural practices, incentives for private investments in low-carbon technologies such as carbon capture and storage infrastructures, and the creation of a materials (including battery) recycling and repurposing ecosystem towards achieving a circular carbon economy. For this to happen, policies and regulations must be aligned to reduce the climate change impacts of the overall energy-technology system. Miscounting carbon, particularly in emission regulation, is a serious error that is fixable with adopting a more comprehensive accounting policy framework.”

Reviewer #2:

Comment 1: I am satisfied with the revisions to the paper.

Response to the Reviewer: We thank the reviewer for taking the time to provide constructive feedback.

Comment 2: However, I would suggest that the authors revise the abstract to remain closer to the findings and avoid broad generalisations e.g., In ln6 , it is not clear as to what BEVs complement. In ln 10 60% increase in central regions however, in the paper, there is no central region.

Response to the Reviewer: The journal word count limits us for the abstract (150 words max), and therefore we have removed this sentence completely and re-written the abstract. See below for the revised abstract (current word count 150 words):

“Battery-electric vehicles (BEV) have emerged as a favoured technology solution to mitigate transport greenhouse gas (GHG) emissions in many non-Annex 1 countries, including India. GHG mitigation potentials of electric 4-wheelers in India depend critically on when and where they are charged: 40% reduction in the north-eastern states and more than 15% increase in the eastern/western regions today, with higher overall GHGs emitted when charged overnight and in the summer. Self-charging gasoline-electric hybrids can lead to 33% GHG reductions, though they haven’t been fully considered a mitigation option in India. Electric 2-wheelers can already enable a 20% reduction in GHG emissions given their small battery size and superior efficiency. India’s electrification plan demands up to 125GWh of annual battery capacities by 2030, nearly 10% of projected worldwide productions. India requires a phased electrification with a near-term focus on 2-wheelers and a clear trajectory to phase-out coal-power for an organised mobility transition.”

Reviewer #3:

Comment 1: I Have gone through the manuscript and it has been substantially improved and is acceptable for publication

Response to the Reviewer: We thank the reviewer for all the constructive feedbacks and suggestions.

Reviewer comments, third review -

Reviewer #1 (Remarks to the Author):

The authors have responded to all of my review comments satisfactorily. The article is now suitable for publication. I commend the authors for their hard work and rigorous analysis.

Reviewer #2 (Remarks to the Author):

I have reviewed earlier and am satisfied with the revisions made